# Debris-flow velocity and volume estimations based on seismic data

Andreas Schimmel[1,2], Velio Coviello[1,3], and Francesco Comiti[1]

[1]Free University of Bozen-Bolzano, Faculty of Science and Technology, Italy
[2]Andreas Schimmel - Alpine Monitoring Systems (ALMOSYS), Mkt. Piesting, Austria
[3]Research Institute for Geo-Hydrological Protection, Consiglio Nazionale delle Ricerche, Padova, Italy

**Correspondence:** Velio Coviello (velio.coviello@cnr.it)

**Abstract.** The estimation of debris-flow velocity and volume is a fundamental task for the development of early warning systems, the design of control structures and of other mitigation measures. Debris-flow velocity can be calculated using seismic data recorded at two monitoring stations located along the channel and previous analysis of the seismic energy produced by debris flows showed that the peak discharge of each surge can be estimated based on the maximum amplitude of the seismic signal. This work provides a first approach for estimating the total volume of debris flows from the integrated seismic energy detected with simple, low-cost geophones installed along a debris-flow channel. The developed methods were applied to seismic data collected from 2014 to 2018 in three different test sites in the European Alps: Gadria and Cancia (Italy), and Lattenbach (Austria). An adaptable cross-correlation time window was used to calculate the velocity of the different surges, which can offer a better estimation of the velocity compared to a constant window length. The analyses of the seismic data of 14 debris flows shows the strong control of the sampling rate and of the inter-stations distance on velocity estimation. A linear relationship between the squares of seismic amplitudes - a proxy for seismic energy - and independent measurements of the debris-flow volume is proposed for a first order estimation the latter. Uncertainties in the volume estimations are controlled by flow properties - such as liquid or viscous surges generating low amplitude signals and thus underestimating the calculated volume - but in most cases (9 out of 11 events of the test dataset of the Illgraben basin, CH) the order of magnitude of the debris-flow volume is correctly predicted.

## 1   Introduction

With the rapid socio-economic development of European mountain areas, the automatic detection and identification of mass movements like landslides, debris flows, and avalanches become of paramount importance for risk mitigation. Technological development has rapidly advanced during the last decade, as well as the conceptual advancements brought by former debris-flow research, making the implementation of monitoring devices for research, early warning and alarm purposes more and more effective (Hürlimann et al., 2019). Past studies showed that such processes induce characteristic seismic and acoustic signals, the latter mostly in the infrasonic spectrum which can thus be used for event detection. Seismic-based monitoring and warning systems have become increasingly applied worldwide to mitigate risks associated to debris flow processes. Several investigations have already addressed signal processing and detection methods based on seismic (e.g., Coviello et al., 2018; Walter et al., 2017; Burtin et al., 2016) or infrasound sensors (e.g., Zhang et al., 2004; Ulivieri et al., 2012; Marchetti et

al., 2019). However, for developing an efficient warning system, not only the detection of events is important but also the identification of the event type (e.g. debris flow vs debris flood as they have different momentum and thus damaging power) and the estimation of its velocity and volume.

An early approach to estimate the process velocity based on seismic data and cross-correlation was proposed by Arattano and Marchi (2005). Later, Havens et al. (2014) and Marchetti et al. (2015) used arrays of infrasound sensors to estimate the velocity of snow avalanches. Differently, Takezawa et al. (2010) developed a method by which flow velocity is estimated based on the amplification rate of the seismic signals of debris flows. The estimation of the debris-flow volume based on seismic data is still an open problem. A quantitative characterization of the event size based on theoretical models (e.g., Lai et al., 2018; Farin et al., 2019) is difficult because of the limited knowledge on the radiated wavefield produced by debris flows and of the uncertainties due to the heterogeneity of the media (Allstadt et al., 2019; Kean et al., 2015). Some possible approaches can be found in the methods used to analyze the seismic signals generated by other processes, such as rockfalls. Manconi et al. (2016) presented an estimation of rockslides volumes based on the ratio between the local magnitude and the duration magnitude detected by broadband seismic networks. The first is computed using the maximum amplitude while the second using the total duration of the seismic train produced by the seismic source (i.e., the rockslide). Controlled experiments point to the relationships among the potential energy lost, the kinetic energy and the radiated seismic energy and allow to retrieve the rockfall mass from the seismic signal (Hibert et al., 2017). Le Roy et al. (2019) found a relation between the potential energy of a free-fall rockfall and the seismic energy generated during the impact that allows to estimate the rockfall volume. For debris flows, Coviello et al. (2019) investigated the energy radiated by natural debris flow surges deducing a scaling relation between kinetic and seismic energy. Interestingly, Pérez-Guillén et al. (2019) deduced similar scaling relationships based on seismic parameters to quantify the size of mass flows at Mt. Fuji, Japan, independently from the type of flow (avalanches or lahars) and from the flow path. Using such scaling relationships, the estimation of the flowing mass is possible based on the seismic energy detected by a geophone and the information about flow velocity. Despite such recent advances, the estimation of debris flow volume from seismic data only is a challenging task in the perspective of the real time event characterization, and uncertainties in the volume estimations are still large (Coviello et al., 2019; Pérez-Guillén et al., 2019; Walsh et al., 2012). Remarkably, most of the (quite few) studies published so far on this topic have addressed estimations in single catchments only.

This paper explores the possibility to develop a simple method to estimate debris flow velocity and volume based on data from seismic sensors installed along the channel, with a limited calibration dataset. The aim is not to seek a universal law relating seismic energy to debris flow characteristics, but just to provide robust tools for debris flow risk management. Specifically, the proposed method is intended to be easily applicable in different catchments, at least for first order estimations of debris-flow volumes.

## 2 Methods

Data collected in three small catchments located in the European Alps prone to frequent debris flows are analysed here: Gadria (South Tyrol, Italy), Cancia (Veneto, Italy) and Lattenbach (Tyrol, Austria). The data of Illgraben (Vallis, Switzerland) is used to test the developed volume estimation methode.

The Gadria basin is located in the Vinschgau-Venosta valley, in South Tyol (Eastern Italian Alps). It has a catchment area of 6.3 km$^2$, ranges in elevation from 2,945 m a.s.l down to 1,394 m a.s.l and is characterized by a regular debris-flow activity. The monitoring system consists of rain gauges, flow stage sensors, geophones, video cameras, piezometers and soil moisture probes. Debris flow depth is monitored by radar sensors installed at three cross-sections along the main channel. A linear array of geophones is used for event detection based on a STA/LTA algorithm (Coviello et al., 2019) and such geophone data can also be used to calculate the velocity. The geophones G1, G2 and G3 used for the calculation of the velocity (marked with a yellow circle) are placed in a distance of 100 m (G1,G2) and 75 m (G2,G3) along the channel. The geophone G4 (marked with a red circle) used for the volume estimation is part of a debris flow detection system based on a combination of infrasound and seismic sensors. This detection system (MAMODIS) consists of one infrasound sensor, one geophone and a microcontroller, where a specially designed detection algorithm is executed to detect events in real time directly at the sensor site (Schimmel and Hübl, 2016; Schimmel et al., 2018).

The Cancia channel is located in the Dolomites within the Province of Belluno(Italy), and the catchment features an area of 2.5 km$^2$ on the southwestern slope of Mount Antelao (3264 m a.s.l.). The catchment ranges in elevation between the Salvella Fork at 2500 m a.s.l. down to a retaining basin at the village of Cancia at 1001 m a.s.l. (Gregoretti et al., 2019). The data used for the volume estimation and velocity calculation are recorded by the geophones installed at station 1 and 2 belonging to the monitoring and warning system designed by the company CAE (CAE, 2014; Cavalli et al., 2020). Geophone G1 and G3 are used for the velocity estimation and geophone G2 is used for the volume estimation. Beside a monitoring system of the company CAE, three monitoring stations have been installed by Universities of Padova, Bologna and Bolzano in 2019. These monitoring stations include two laser stage sensors, two rain gauges, several time-lapse cameras, geophones and the infrasound/seismic detection system MAMODIS and integrates a monitoring network that was operational in the previous years only for scientific purposes (Simoni et al., 2020).

Finally, the Lattenbach Creek (district of Landeck, Tyrol) has a catchment area of 5.3 km$^2$ and is a monitoring site for debris flows operated by the Institute of Mountain Risk Engineering at the University of Natural Resources and Life Sciences, Vienna (Hübl and Moser, 2006). Three monitoring stations are installed along the channel, and these are equipped with flow height (radar gauges), geophones, video cameras, 2D-Laser scanner. At the middle monitoring station, a debris flow Pulse-Doppler Radar can be used for measuring the surface velocity. Near this radar, two stations for testing the warning system MAMODIS are installed at a distance of 90 m. The geophone data from the two stations (G1 and G2) are used to calculate the debris flow velocity and the lower one (G2) is used for the volume estimation in this study. Figure 1 gives an overview of the three catchments and the monitoring setup.

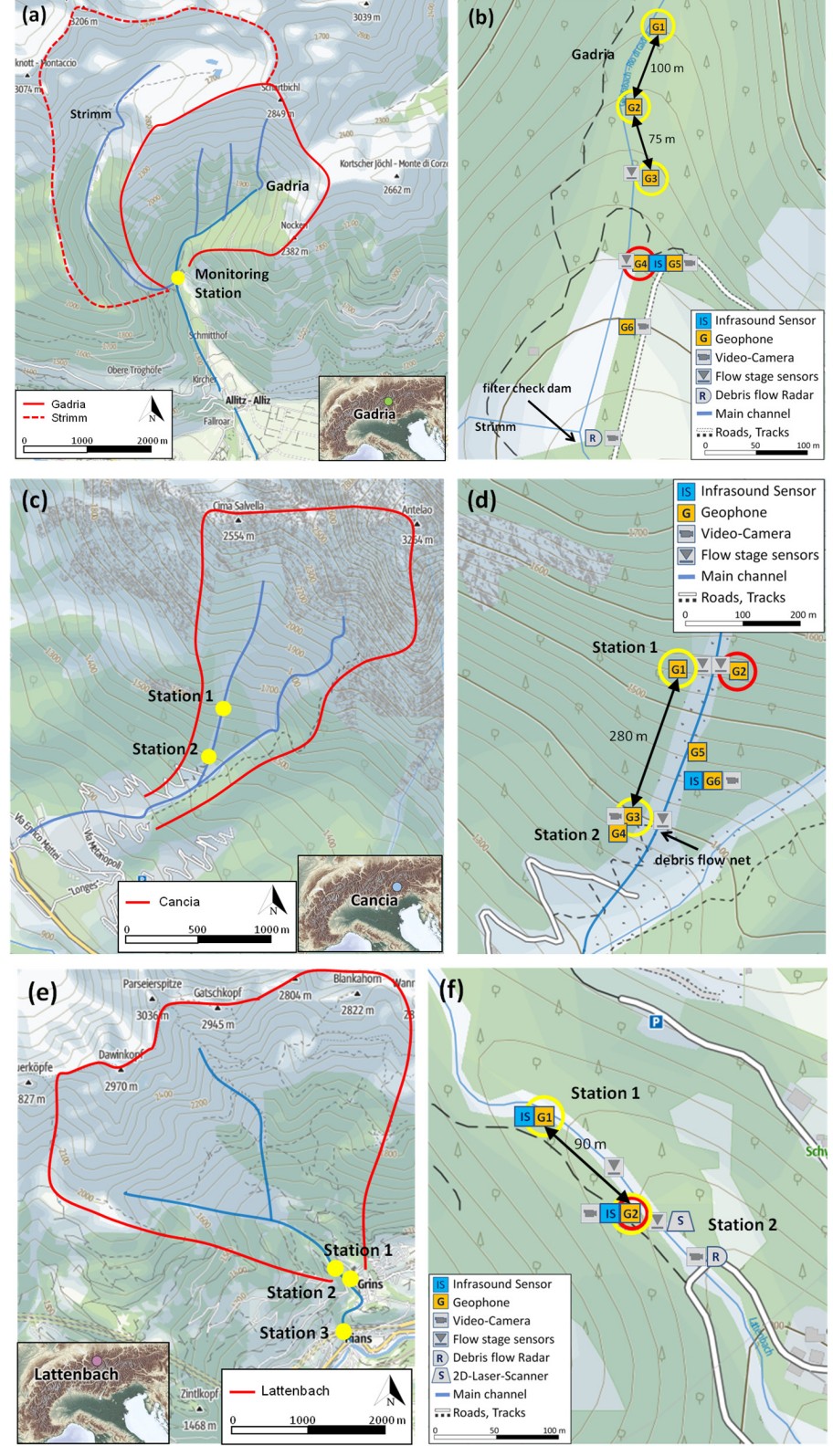

**Figure 1.** Overview of the Gadria site (a); Cancia site (c); Lattenbach site (e); (b),(d),(f) Closer view of the monitoring stations and sensor setup (based on ©OpenStreetMap-contributors)

**Table 1.** Summary of the seismic monitoring equipment

| | Geophone | Type | Natural freq. [Hz] | Sensitivity [Vsm$^{-1}$] | Sampling rate [Hz] | Amp. values [Hz] |
|---|---|---|---|---|---|---|
| Lattenbach | G1/2 | Sercel SG-5 | 5 | 80 | 100 | 1 |
| Gadria | G1/2/3 | Geospace | 10 | 85.8 | 128 | 0.5 |
| | G4 | Sercel SG-5 | 5 | 80 | 100 | 1 |
| | G5/6 | Sensor NL SM-6 | 4.5 | 28.8 | 100 | 1 |
| Cancia | G1/2/3/4 | SolGeo VELOGET-DNL-H | 14 | 18.2 | 500 | 0.1 |
| | G5/6 | Sensor NL SM-6 | 4.5 | 28.8 | 100 | 1 |

Table 1 gives an overview of the seismic sensors used at the different sites. The seismic amplitudes used for this study are
90 calculated every second as signal envelope (Arattano et al., 2014) from the raw data recorded at the reported sampling rates.
At Cancia, an internal sampling rate of 500 Hz is used, but the available seismic data are recorded as 0.1 Hz max. amplitude
values. For the geophones of the type SG-5 and SM-6 amplitude values of 1 Hz are calculated from the raw signals sampled at
100 Hz and at Gadria the used data for this study are 0.5 Hz amplitude values.

## 2.1 Velocity estimation

The estimation of debris flow velocity is carried out by the time-distance method, whereby velocity is calculated as the dis-
tance between two stations measuring seismic amplitude along the channel divided by the time difference of the two signals
calculated from amplitude maximum values (Coviello et al., 2021; Schimmel et al., 2018), or by cross-correlation of the two
seismic signals (Arattano et al., 2012). The result of this method is a mean surge velocity (celerity) between the two stations.
To obtain the time difference based on amplitude maxima, the signal is manually analysed identifying comparable peaks (i.e.,
representing the debris flow front or subsequent surges) in the signals recoded at the two stations. The manual analysis is used
for validating the results of application of the cross-correlation method. For the cross-correlation analysis, the analysis window
size has to be selected. After testing several settings, we decided to use a starting window size related to the distance of the two
geophones. This choice offers the best result for the cross-correlation and provides an objective method, based on one parame-
ter (distance) only, to adapt the cross-correlation analysis at new sites. The number of samples is set equal to the distance in m,
which means that a resolution from $1\ \mathrm{ms^{-1}}$ is possible which seems to be a phyisically meaningful starting value for describing
turbulent debris flows (e.g., Theule et al., 2018). Three different sliding time window sizes are used because an adaptation of
the time window ensures better results for the cross-correlation for all flow stages. For choosing the window length, the ratio
between maximum amplitude and minimum amplitude is analysed in the starting window size which has a number of samples
equal to the distance. Analyses of the seismic data of twelve events (3 at Gadria, 3 at Cancia and 6 at Lattenbach) showed that
when such ratio >6 the debris flow features an adequate signal shape for cross-correlation to be adopted. If the ratio is <6, the

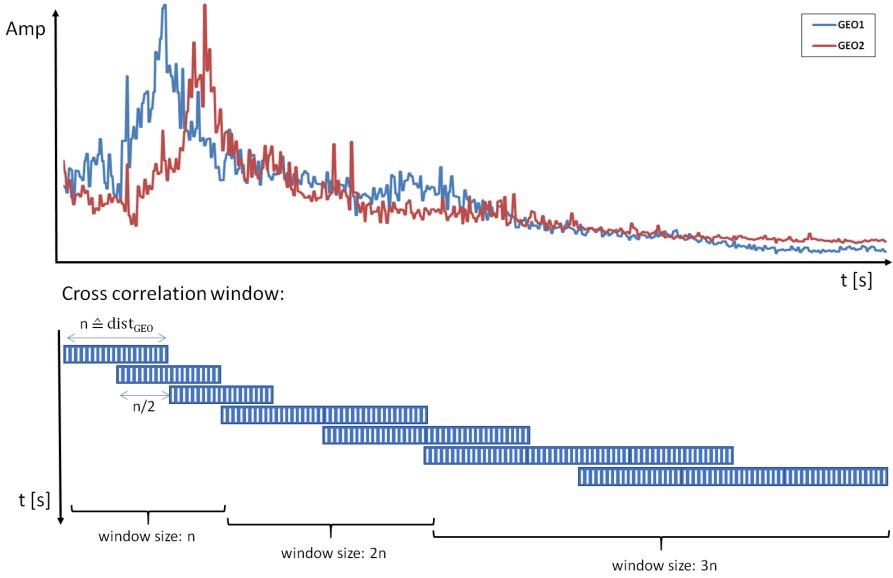

**Figure 2.** Methode cross-correlation analysis: window sizes and overlaps.

window length will be enlarged by another number of samples equal to the distance. If the signal shape still is not suitable, the window will be further expanded. Figure 2 shows the principle of the adaptive window sizes. Therefore, the lowest velocity that can be calculated is theoretically $1 \text{ ms}^{-1}$ in the first, typically rougher part of the debris flow hydrograph, with a signal length equal the distance, and it could reduce to $0.33 \text{ ms}^{-1}$ for the smoother, tail phase of the event if a window length of three times the distance is used. Cross-correlation is performed with an overlap of half of the sample numbers. The two signals are normalized in the window frame by the maximum amplitude value. Only if the cross-correlation coefficient exceeds 0.8 the result is kept for the velocity calculation. This threshold was selected by a trial and error procedure. Since the cross-correlation analyses is performed at $1 \text{ s}$ time steps, the Cancia and Gadria data are upsampled to a sample rate of $1 \text{ Hz}$. Therefore amplitude values from Cancia and Gadria are constant over $10 \text{ s}$ and $2 \text{ s}$, respectively.

## 2.2 Volume estimation

As reported in the introduction, a linear trend between the seismic energy (J), which is proportional to the square of the seismic amplitude $(\text{m}^2\text{s}^{-2})$, and the kinetic energy per unit area produced by debris flows has been observed by Coviello et al. (2019), Consequently, we integrated the squared amplitude values during the whole duration of a debris flow to obtain an estimation of the seismic energy of each event. To make the results comparable for all three sites and not depending on different detection methods, the used event duration has been determined manually based on the signal shape. Subsequently, we related these integrals of the seismic signal to the associated debris flow volumes. For these latter, we used published and unpublished estimates obtained by several methods (topographic surveys, stage sensors, 2D scanners and debris flow radar) in the study basins (Schimmel et al., 2018; Coviello et al., 2021; Simoni et al., 2020). Overall, a total of 14 events (occurred from 2014

to 2018) are available from the three different catchments (Table 2). The best fit curve relating debris flow volumes to the seismic signal was obtained by performing a linear regression analysis. The performances of the methods are investigated and discussed using 11 independent debris-flow volumes recorded at Illgraben, Switzerland, from 2015 to 2017 (Schimmel et al., 2018; Marchetti et al., 2019). Since all monitoring stations used for this study are rather close to the channel (between 15 and 25 m) and the distances are nearly the same at every test site, attenuation of the signals in the ground, geometric spreading and the influence of topography or geology can be neglected. Also intense rainfall and wind can produce ground vibration that geophones detect. However, seismic signals recorded by sensors installed at small distance from the channel (from 15 to 25 meters, in our case) are dominated by in-channel processes. This is particularly true in our study sites, which are located in lower reaches of the main channels where the debris flow surges are well formed and characterized by velocities of several meters per seconds and flow depth in the order of meters. To make data analysis comparable among the sites, the lowest sampling rate (10 s for the Cancia dataset) is used, and seismic data from the other catchments are transformed in terms of maximum values of amplitude over periods of 10 s.

**Table 2.** List of event dates and volumes for all sites. Data gathered in Gadria, Cancia and Lattenbach were used to retreive the empirical equation 1 while data from Illgraben for validation, see Figure 7.

|  | Date | tot. Volume [$m^3$] | Duration [s] | Reference |
|---|---|---|---|---|
| Lattenbach | 09.08.15 | 11500 | 1600 | Schimmel et al. (2018) |
|  | 10.08.15 | 18500 | 2800 | Schimmel et al. (2018) |
|  | 16.08.15 | 5000 | 1200 | Schimmel et al. (2018) |
|  | 10.09.16 | 46000 | 3900 | Schimmel et al. (2018) |
|  | 29.07.17 | 14000 | 1600 | Internal report (2018) |
|  | 30.07.17 | 41000 | 3500 | Internal report (2018) |
| Gadria | 15.07.14 | 11600 | 2000 | Coviello et al. (2021) |
|  | 08.06.15 | 12600 | 3300 | Coviello et al. (2021) |
|  | 12.07.16 | 2400 | 2500 | Coviello et al. (2021) |
|  | 19.08.17 | 2300 | 1400 | Coviello et al. (2021) |
| Cancia | 23.07.15 | 25000 | 1600 | Simoni et al. (2020) |
|  | 04.08.15 | 20000 | 2000 | Simoni et al. (2020) |
|  | 01.08.18 | 4500 | 2700 | Simoni et al. (2020) |
|  | 29.10.18 | 11000 | 3200 | Simoni et al. (2020) |
| Illgraben | 22.07.15 | 8700 | 3500 | Schimmel et al. (2018) |
|  | 10.08.15 | 6100 | 6700 | Schimmel et al. (2018) |
|  | 14.08.15 | 25000 | 9500 | Schimmel et al. (2018) |
|  | 15.08.15 | 2000 | 5500 | Schimmel et al. (2018) |
|  | 12.07.16 | 10000 | 4200 | Schimmel et al. (2018) |
|  | 12.07.16 | 60000 | 3000 | Schimmel et al. (2018) |
|  | 22.07.16 | >10000 | 3000 | Schimmel et al. (2018) |
|  | 09.08.16 | <10000 | 2500 | Schimmel et al. (2018) |
|  | 29.05.17 | 70000 | 3500 | Marchetti et al. (2019) |
|  | 04.06.17 | 24000 | 2800 | Marchetti et al. (2019) |
|  | 14.06.17 | 33000 | 3100 | Marchetti et al. (2019) |

## 3 Results

First we present the results about velocity estimation adopting the methods described above, applied to three debris flows events recorded in different catchments. Figure 3 illustrates velocity estimations applied to the Lattenbach event occurred on 30 July 2017, which featured a peak discharge of 88 $\mathrm{m^3s^{-1}}$, a total volume of 41,100 $\mathrm{m^3}$ and an overall duration of around 3500 s. This debris flow had a front about 1.3 m high, and the velocity (3.5 to 4.7 $\mathrm{ms^{-1}}$) calculated by using the time difference between maximum amplitude values results very similar to the velocity calculated by cross-correlation with 4 $\mathrm{ms^{-1}}$. For the peak discharge (flow height exceeding 3.5 m, the velocity calculated by means of maximum values turns out slightly higher (10 $\mathrm{ms^{-1}}$) than the one (9 $\mathrm{ms^{-1}}$) determined by cross-correlation. During the following part of the event (i.e., after 2500 s) no significant surges could be found to calculate flow velocities using maximum values, and the cross-correlation most likely leads to overestimating velocities due to such lack of surges.

Figure 4 displays the seismic signals and the velocity estimation for a debris flow occurred in the Gadria on 08 June 2015, which was characterized by a total volume of 12,600 $\mathrm{m^3}$. The event is composed of several surges in the range 1-1.5 m of flow height. The front velocity and the velocity of the surge visible at 2000 s seems to be overestimated by the cross-correlation method, because velocities over 9 $\mathrm{ms^{-1}}$ and 7 $\mathrm{ms^{-1}}$ respectively seem unrealistically high based on previous results from the Gadria (Theule et al., 2018; Coviello et al., 2021). In contrast, for the other surges, flow velocities calculated based on maximum values and cross-correlation give consistent estimates, around 5 $\mathrm{ms^{-1}}$.

Finally, Figure 5 shows the case of a debris flow in the Cancia channel. This event was recorded on 01 July 2020. While the debris flows height reaches 2.4 m, flow velocities for this event appear to be lower (max. 3.2 $\mathrm{ms^{-1}}$) than in the case of Lattenbach and Gadria.

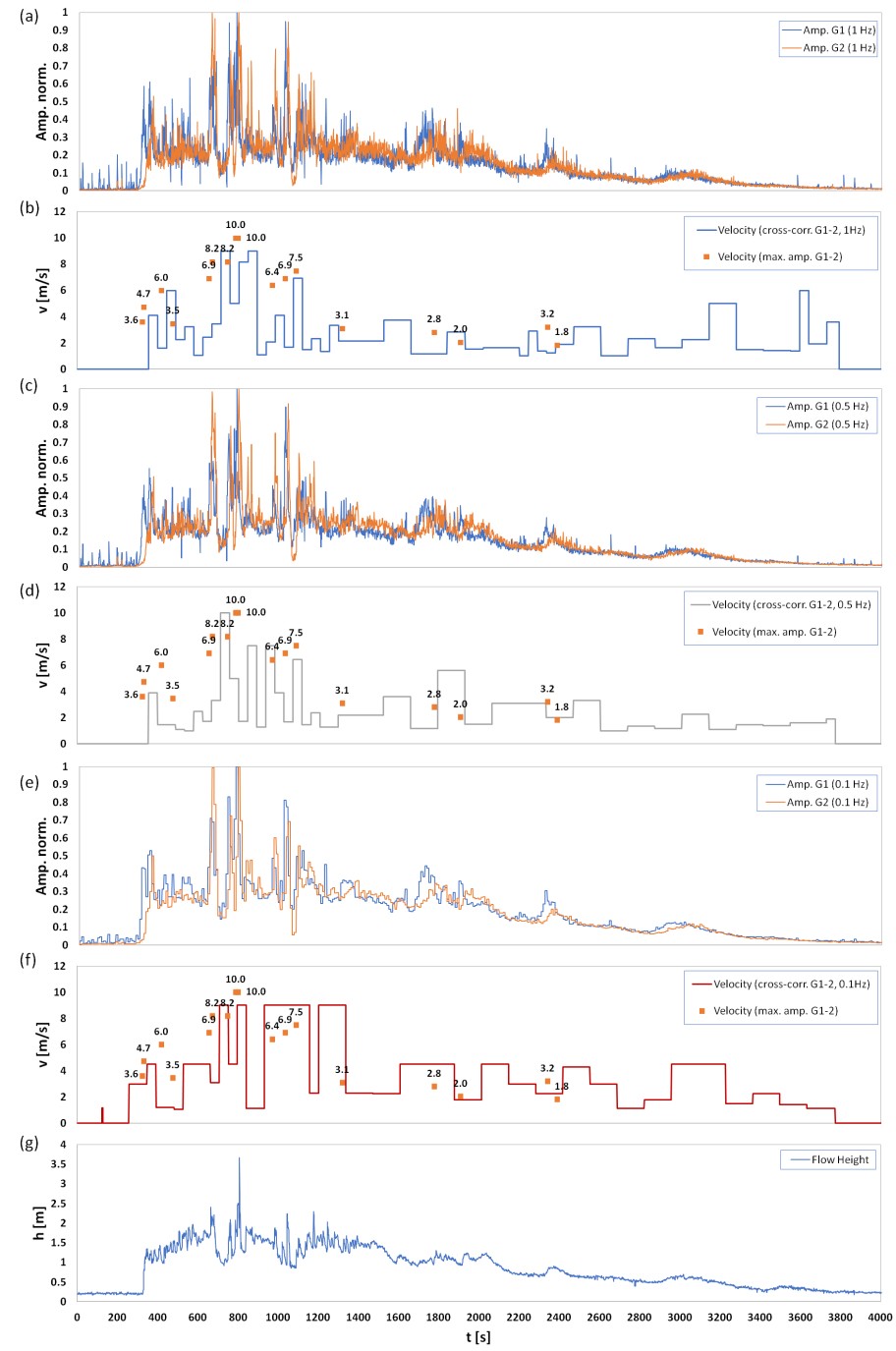

**Figure 3.** Debris flow at Lattenbach on 30 July 2017: (a,c,e) normalized amplitudes of the two geophones (G1,G2), (g) flow height, (b,d,e) velocity estimation based on maximum values and cross-correlation (compared for sampling rates of 1 Hz, 0.5 Hz and 0.1 Hz)

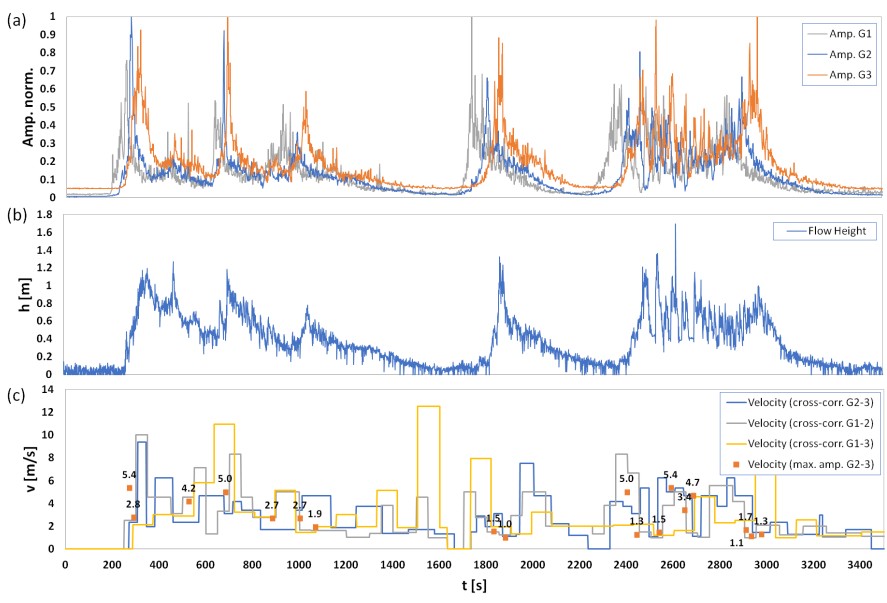

**Figure 4.** Debris flow at Gadria on 08 June 2015: (a) Normalized amplitudes of the three geophones (G1,G2,G3), (b) flow height, (c) velocity estimation based on maximum values and cross-correlation

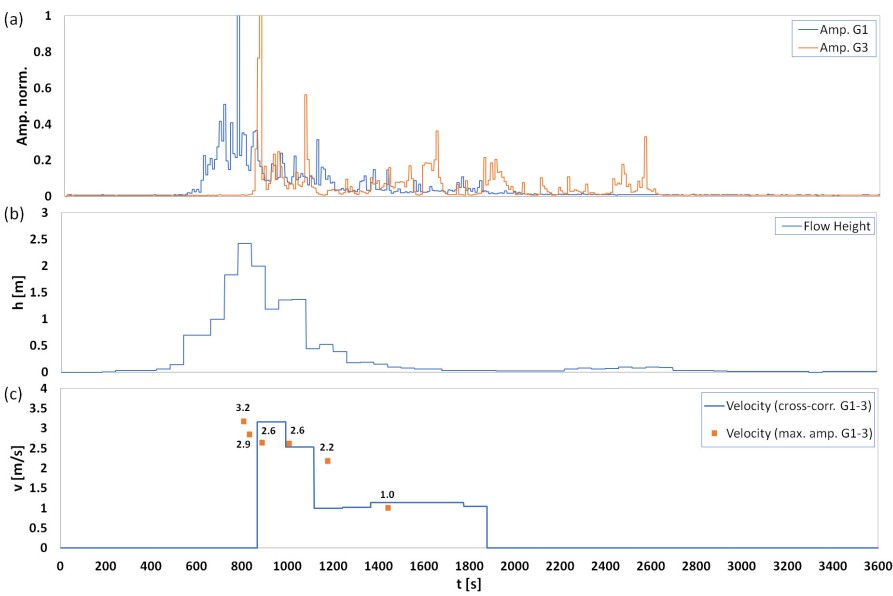

**Figure 5.** Debris flow at Cancia on 01 July 2020: (a) normalized amplitudes of the two geophones (G1,G3), (b) flow height, (c) velocity estimation based on maximum values and cross-correlation

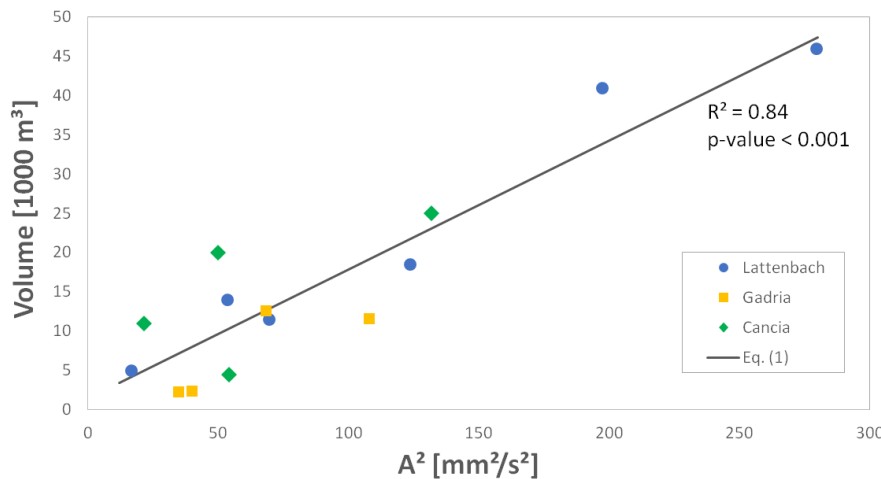

**Figure 6.** Relationship between squared integrated seismic amplitude and total volume based of the debris flow events listed in Table 2 (from Gadria, Lattenbach and Cancia)

To test the methodology described above for the estimation of debris flow volumes based on seismic signals, a total of 14 events (occurred from 2014 to 2018 ) are available from the three different catchments (Table 2). Figures 6 shows that the use of the squared seismic amplitudes ($A^2$ in $\mathrm{mm^2 s^{-2}}$) with a linear fitting seems most promising to provide a preliminary estimate of event volumes ($V_{tot}$ in $\mathrm{m^3}$) compared to other curve fitting approaches like power law ($R^2 = 0.56$) and exponential fitting ($R^2 = 0.57$). The best fitting linear equation reads:

$V_{tot} = 164\,A^2 + 1419$                          (1)

    The method has then been tested against 11 independent debris-flow volumes recorded at Illgraben, Switzerland. Figure 7 compares all the observed values (vertical axis) for total volume to the predicted values (horizontal axis) according to Eq. 1. Two events at Illgraben plot quite far off the confidence level shown in Figure 7. Possible reasons for the poor prediction of their volumes by Eq. 1 is provided in the discussions.

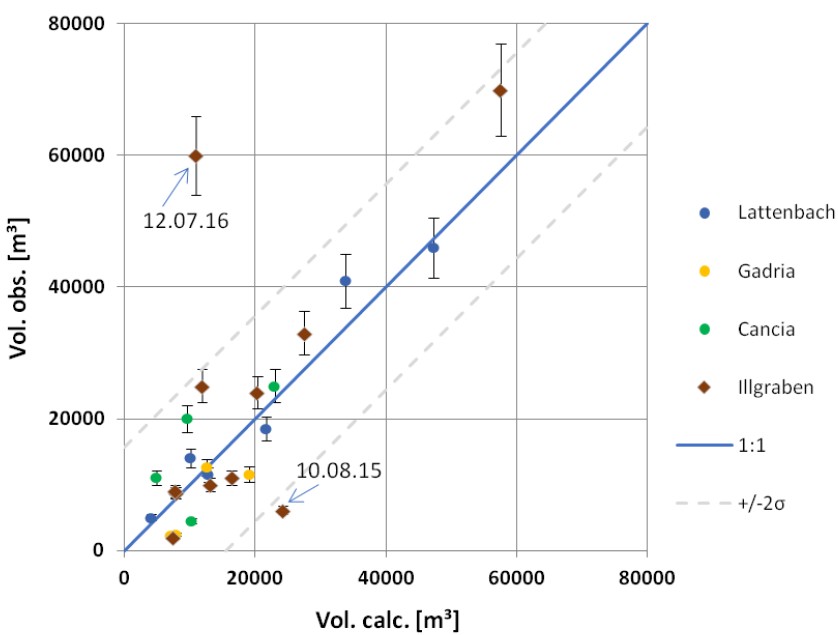

**Figure 7.** Comparison of the predicted volume vs observed volume. The dark blue line represents the one-to-one relationship and the dashed lines represent the confidence interval of the distribution.

## 4  Discussion

Our results suggest that the cross-correlation method we used - based on a window length adaptable according to the signal waveform - provides solid estimates of debris-flow velocity, as the temporal resolution of the calculation is high during the fast, initial stages of the flow, while longer window length are applied for smoother flows, thus permitting to avoid wrong correlation results. Arattano et al. (2012) already showed how the cross-correlation technique can provide a reliable estimation
of the flow velocity even when the signals recorded at the two monitored cross-sections do not present a clear, common feature, i.e., typically the passage of the debris-flow front. Nonetheless, some significant signal features are required such as a progressive rise and subsequent decrease of the signal amplitude. Signals characterized by many amplitude peaks close to each other produced, for instance, by the propagation of roll waves can represent a limitation to the application of cross-correlation methods (Figure 5c). The validation approach for the velocity estimates, i.e. manually determining matching amplitude peaks
at the two stations, is also affected by uncertainty. In the Gadria, this is particularly evident for the tail of the debris flow of 08 June 2015 (Figure 4c, from t = 2400 s) during which roll waves propagate and produce multiple peaks, one following the other. The uncertainty in the manual velocity calculation was also observed in previous analysis based on data gathered with a pair of flow stage sensors and led to the approximation of lumping multiple waves into one single surge for the subsequent volume estimation (Coviello et al., 2021). The velocity estimates of surges lacking multiple peaks (i.e., from t = 200 s to 2400 s in
Figure 4c) are consistent with those performed with the flow stage sensors located downstream from G3 (Figure 1c). Indeed, they are slightly higher (i.e., differences from 0.3 to 1.7 $\mathrm{ms}^{-1}$) than those calculated by the flow stage data on a milder sloping channel reach (Coviello et al., 2021).

Importantly, our study benefited from three, quite different test sites. The influence of different distances between the geophones is evident on the velocity estimation. The longitudinal geophone distance in the Gadria (75 m) and Lattenbach (90 m)
appear to be appropriate for fast debris flows, while the longer distance in Cancia (280 m) makes difficult or even impossible capturing the same surges at different sensors. In any case, the transversal distance between the channel and the geophones should be much smaller (at least the half) than the longitudinal distance between the two geophones (Coviello et al., 2019). The distance has to be chosen to provide a significant difference in the signals in an appropriate time, so that the cross-correlation offers valid results for flow velocity.
The sampling rate also has an important effect on the reliability of velocity estimations. At Lattenbach and Gadria, one amplitude value every 1-2 s was available. This seems to be a proper sampling in combination with the sensor distances. At Cancia, only one sample every 10 s is available, so that the signal shapes can be very different at the two geophones, determining problems for the cross-correlation analysis. In fact, surges can be missed and such a low sampling rate coupled with the long distance lead to an exaggerated averaging of flow velocity of different surges. This might has an effect on the
calculated velocity values in Cancia, which are much lower compared to the other sites. However, in Cancia velocities estimated on the basis of image analysis of time-lapse videos on previous events (Simoni et al., 2020) are in the same range ( e.g. 1.5 to 4 $\mathrm{ms}^{-1}$ for a debris flow on 23 July 2015). Therefore we believe that the lower velocities in Cancia compared to Gadria and

Lattenbach stem for the different characteristcs of debris flows of this catchment, which are more granular compared to the other sites.

We performed a test on the debris-flow event recorded at Lattenbach on 30 July 2017 (Figure 3). Seismic data of this event were recorded at 1 Hz. We subsampled data at at 0.5 and 0.1 Hz and we compared the flow velocity calculated on these three signals. Figure 3 shows remarkable differences when adopting the cross-correlation technique at different sampling rate. Apart from the obiviously larger duration of the time windows, the signal subsampled at 0.1 Hz produces an overestimation of the flow velocity of the main surges (i.e., from t = 500 to t = 1500 s) compared to the original signal.

Different sensors other than the geophones can be used to determine debris flow velocity. So instead of geophones two separated stage sensors can be used for the time-distance method. The advantage of stage sensors is that they measure the process directly, so there are no effects of ground damping, channel texture or the viscosity of the process, which have a high influence on the seismic signal shape. On the other side, stage sensors need a structure above the channel, so they have a much higher installation effort and are more exposed to the debris flow (Coviello et al., 2019). Alternatively, flow velocity can be measured by Pulse Doppler radar (Koschuch et al., 2015). This method calculates the velocity from the frequency shift of a pulse-modulated high-frequency reflected radar signal, which is proportional to the velocity of the moving object (Doppler effect). The detection area is divided in different range gates and the result is an instantaneous surface velocity distribution (velocity spectrum) for each range gate. Therefore, a debris flow radar measures the velocity directly, but there is an averaging over the range gate, so the surge velocity measured by the radar is often lower than the surge velocity measured by the time-distance method. When velocity data measured by the debris flow radar in the Lattenbach (unpublished data) are compared against values calculated from the geophones there installed, very similar results can be observed. In fact, the maximum velocity measured by the debris flow radar for the event on 30 July 2017 is $10.0 \ \mathrm{ms}^{-1}$, while the maximum value calculated from the geophone data is $9.0 \ \mathrm{ms}^{-1}$. The mean velocity of the whole event is $1.8 \ \mathrm{ms}^{-1}$ based on the debris flow radar, and $1.9 \ \mathrm{ms}^{-1}$ for the presented method based on the geophone data.

A linear trend between the square of the seismic amplitudes and the debris flow volumes is apparent from analysis conducted by merging the three sites. The fact that a linear model performs definitely well is in agreement with the physical processes linking seismic energy to debris flow parameters such as mass and velocity combined or peak discharge, as already noted by other authors (Coviello et al., 2019; Andrade et al., 2022). Figure 7 compares the observed values (vertical axis) of total volume to the predicted values (horizontal axis) of all the debris flow events reported in Table 2. Data gathered at Gadria, Cancia and Lattenbach represent the test dataset while the validation dataset is composed of debris flows observed in the Illgraben catchment, Switzerland, from 2015 to 2017 (Table 3). This analysis suggest that it is possible to obtain first-order estimates of debris flow volumes based on the seismic amplitudes, but there is still a large variance, since there are several factors affecting the seismic signals: distance geophone - channel, damping in the ground or sampling rate (e.g., Kean et al., 2015; Coviello et al., 2018; Allstadt et al., 2019). As already highlighted in the results, two events in the Illgraben out of eleven that compose the validation dataset (debris flows observed at Illgraben) plot out of the confidence interval of the distribution (2 $\sigma$). The error in the volume prediction of the 10 August 2015 event is possibly due to the significantly higher velocity of this event compared to the others (Schimmel et al., 2018). Indeed, the volume prediction is strongly controlled by the velocity

**Table 3.** Errors in the volume prediction for the Illgraben test dataset.

| Event date | Observed Vol. [m$^3$] | Predicted Vol. with Eq. 1 [m$^3$] | Error Eq. 1 | Predicted Vol. with scaled $A$ [m$^3$] | Error scaled $A$ | Notes |
|---|---|---|---|---|---|---|
| 22.07.15 | 8700 | 7866 | 10% | 8167 | 6% | |
| 10.08.15 | 6100 | 24208 | -297% | 24667 | -304% | high flow velocity |
| 14.08.15 | 25000 | 11907 | 52% | 12247 | 51% | |
| 15.08.15 | 2000 | 7503 | -275% | 7800 | -290% | smallest event |
| 12.07.16 | 10000 | 13148 | -31% | 13500 | -35% | |
| 12.07.16 | 60000 | 10933 | 82% | 11263 | 81% | liquid front, viscous tail |
| 22.07.16 | 11000 | 16414 | -49% | 16798 | -53% | |
| 09.08.16 | 9000 | 7739 | 14% | 8038 | 11% | |
| 29.05.17 | 70000 | 57544 | 18% | 58324 | 17% | |
| 04.06.17 | 24000 | 20344 | 15% | 20765 | 13% | |
| 14.06.17 | 33000 | 27498 | 17% | 27988 | 15% | |

and the mass (i.e., solid content) of the mixture (Coviello et al., 2019). Concerning the other outlier (12 July 2016 debris flow), the velocity of the first surge was high (7.8 ms$^{-1}$) but in the video recording the first part of the flow appears very liquid and the tail viscous. This can explains the low amplitudes of the geophone signal that generate such a small volume when using equation 1. Additionally, the total volume is estimated over the event duration and for an automatic volume estimation (like the method presented in Schimmel et al. (2018)) such event duration is defined by the detection method itself. For example, the amplitude thresholds for the detection criteria has also an influence on the event duration and thus on the total volume estimation.

Nonetheless, adopting such a phyisically-sound empirical model, a near-real time estimate of debris flow surges is possible. However, this volume estimation becomes available only at the end of the surge. This means that the final volume estimation would be provided too late to inform civil protection managers about the flow volume. Therefore, this method is still quite far from the goal of having a real time, accurate volume estimation to be implemented in early warning systems. Nonetheless, a rapid estimate of the order of magnitude of the debris-flow volume would become available, which could be used by local authorities for managing the debris flow event, e.g., by organizing clearing of retention basins, bridges and roads.

We highlight again that our methods is based on seismic data gathered in the near field, i.e. geophone stations located along the channel. For such a volume estimation, small differences in the distance sensor-channel are negligible compared to uncertainties descending from the variability of flow properties. Indeed, the distances sensor-channel for the different sites are

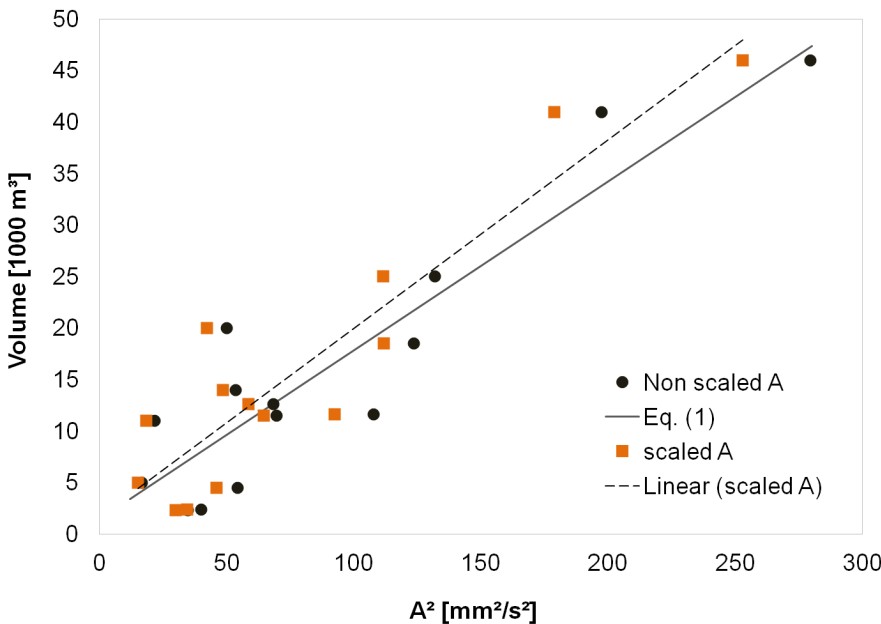

**Figure 8.** Comparison of the relationship between debris flow volumes vs scaled and non-scaled seismic amplitutes.

15 m at Lattenbach, 23 m at Gadria, 25 m at Cancia and 15 m at Illgraben. We then applied a simple empirical relation to
model the decay of the seismic amplitude with distance:

$$A(d) = A\,e^{-\pi\,f\,d\,/\,Q\,v_u} \tag{2}$$

where $d$ is the distance sensor-channel in m. We used a value of quality factor of $Q = 20$ suggested as a reasonable approx-
imation for the relatively high frequencies and shallow depths of interest (Tsai et al., 2012) and a reference value for group
velocity ($v_u$) of 1300 $\mathrm{ms^{-1}}$ (Coviello et al., 2019). An additional approximation was needed to apply the formula using ag-
gregated values of amplitude recorded with different sampling rate and recording frequency (Table 1). We assumed that fronts
of the different surges are the dominant sources of the seismic signal during the respective time window. This assumption is
consistent with our approach of calculating the mean velocity of each surge by means of the cross-correlation technique, which
needs to split the entire debris-flow signal. We tested values of $f$ ranging from 10 to 20 Hz, typical of the main frequency of
debris flows. Higher differences in the scaled amplitude are obtained with the lower frequency value ($f = 20$ Hz) so we used
this latter value in the calculation to maximize the uncertainties. Finally, we recalculated the debris-flow volumes using a linear
regression equation similar to Eq. 1 but based on the square of the scaled amplitudes (Figure 8). Results clearly show how the
differences in the calculated volumes with the non-scaled and the scaled amplitude equations are negligible (Table 3).

Studies of different events also showed a large dependency of the seismic amplitudes and their frequency spectrum on the
velocity of the debris flow. For example, Lai et al. (2018) proposed a model where the seismic amplitudes are most sensitive

to the product of four physical parameters related to the debris flow: length and width of the boulder snout, grain size cubed, and average speed cubed. This model and also the model presented by Farin et al. (2019) shows that a method including the estimation of the debris flow velocity and grain size distribution can result in a more accurate calculation of debris flow volume. The influence of the sediment concentration on the seismic data can therefore improve the results of the volume estimation, but there is still no method to automatically estimate the sediment concentration on seismic data, which could be implemented in the volume estimation. Currently it is only possible to differ between debris flow and debris floods based on the infrasound or seismic peak frequencies (e.g., Hübl et al., 2013), but this still poses large uncertainties and is far from providing reliable estimation of sediment concentration.

## 5   Conclusions

This work shows that important differences can be observed in the debris flow velocity estimation among the different sensor setups at the different catchments. The optimal distance between the sensors, the best sample rate for cross-correlation, or the analysed frequency range has an important influence of the quality of the results. The presented approach with a cross-correlation window length adapted to the signal waveform improves velocity estimation over the entire debris flow duration (from fast, initial stages to smoother flows).

The estimation of the debris-flow volume based on seismic data is still an open problem as theoretical models are still affected by large uncertainties. Starting from the relation between kinetic and seismic energy, our results show that the order of magnitude of debris flow volumes can be correctly estimated from seismic data only, by adopting a linear model based on the squares of the seismic amplitude. However, improvements are necessary for an automatic volume identification usable for a warning system. In fact, flow velocity and the sediment concentration have also a large influence on the seismic amplitudes of a debris flow, so including them in the volume estimation could lead to more accurate results..

*Data availability.* The debris flow waveforms gathered at Gadria and Lattenbach are available through the Exotic Seismic Events Catalog (https://ds.iris.edu/ds/products/esec/)

*Author contributions.* A.S. proposed the idea and analyzed the data. V.C. contributed to the paper writing, to the figure preparation and provided data from the Gadria test site. F.C. supervised the research work. All authors participated in the organization and discussion of results.

*Competing interests.* The authors declare no conflict of interest.

*Acknowledgements.* We thank Johannes Hübl (Institute of Mountain Risk Engineering, University of Natural Resources and Life Sciences, Vienna) for contributing data from the Lattenbach test site, Pierpaolo Macconi (Civil Protection Agency, Autonomous Province of Bozen/Bolzano) for providing data collected by the Gadria station, and Matteo Cesca (Regional Department for Land Safety, Hydrogeological Services Center, ARPA Veneto) for data relative to the Cancia site. Lorenzo Marchi, Marco Cavalli and Stefano Crema (CNR IRPI Padova) are acknowledged for debris flow volumes estimations in the Gadria basin. Andreas Schimmel was funded thanks to "AC-CORDO DI COLLABORAZIONE TECNICO-SCIENTIFICA ex ART. 15 DELLA L.241/90 e s.m.i. PER LA DEFINIZIONE DELLE SOGLIE DI ALLARME E LE CONSEGUENTI LOGICHE DI FUNZIONAMENTO DEL SISTEMA DI MONITORAGGIO E ALLARME DELLA COLATA DETRITICA DI CANCIA IN BORCA DI CADORE)" led by CNR-IRPI Padova in collaboration with Free University of Bozen-Bolzano and the University of Padova and by "Niederösterreichischen Wirtschafts- und Tourismusfonds", F&E project "Automatische Identifikation alpiner Massenbewegungen" (WST3-F-5033340/001-2020). The authors thank the Department of Innovation, Research and University of the Autonomous Province of Bozen/Bolzano for covering the Open Access publication costs.

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
