# Peer review of "Debris-flow velocity and volume estimations based on seismic data"

_Natural Hazards and Earth System Sciences, 2020_

## Author Response (AR1)

**review 1:**

The manuscript presents a study on key debris flow characteristics (velocity and volume) from three study sites. The flow characteristics are estimated using a predominantly seismic approach. The goal of the study is timely, relevant and matches the scope of the journal well. From my perspective, the study design is well suited and the gathered data is substantial, providing proper substance to pursue most of the discussed topics. That said, the manuscript also shows a few drawbacks that need to be addressed before it has reached a state that renders it eligible for publication. I believe all these points can easily be addressed with the data that is inherent to the study.

In the scope section (lines 40-42) the reader gets the impression that the approach would not require any site-specific calibration efforts. This is not true. Evidently, the autors record independent data (flow stage at all four sites, and velocity by a Doppler Radar sensor at one site) and use that (table 2, fig. 8) to convert the integrals of squared seismic velocity to volume (integral of flow stage throughout an event), the corresponding fit function is presented as Eq 1. So, there is a need for calibration.

Yes, we used a site-specific calibration based on complementary data to derive equation (1). However, the aim of this study is to propose a simple method based on few metrics of the seismic signal to obtain a rough estimation of debris-flow volumes. We tested equation (1) against independent debris-flow volumes collected at Illgraben, Switzerland, from 2015 to 2017. The results show that the method offers a good estimation for the volume on an independent test site without additional calibration. We will add the results of this test in the revised manuscript.

Now, it is a matter of discussion whether the authors have demonstrated that this fit function is a universal law to reach suitable results or not. My visual impression of the content of fig. 8 is: No, this is not the case. If one would fit the data of the three sites (coloured symbols) individually, the resulting regression coefficients would be quite different, and that difference could and should actually be tested quantitatively. Hence, when propagating the impact of these supposedly three different regression coefficients to the results shown in fig. 9, I would assume that we would see for each subset of dots quite different results of predicted volumes. In any way, to make things short here: if the authors wish to claim there is a universal law to relate seismic energy integrals to total flow volumes then they have to prove this hypothesis in a proper way.

The aim of our study is to find a simple method to provide an approximate volume estimation that can be implemented in a detection system. By no means we believe that, with such a limited dataset, we have found a universal law relating seismic energy to debris-flow volumes in any conditions. In particular, the higher range of the volume distribution of our dataset is poorly represented, and by using three different regression equations, one per sites, the results would not be statistically significant. In addition, differences in rheology and flow dynamics can have an impact on results. We will better clarify the aims of our study, as well as we will expand the discussion of our results using the validation dataset from the Illgraben mentioned above.

It remained unclear to me, which stations the authors used to calculate the integral of squared ground velocity, one sensor (which one) or all available? Was an average value used (and can the scatter be quantified), or have the amplitudes been scaled by distance to source?

The geophones used for the volume estimation are G4 at Gadria, G2 at Lattenbach, and G2 at Cancia. This information is given in the text describing the test sites, we will clarify this point in the methods section 2.2 Magnitude estimation. At each test sites, we selected the geophone that recorded the larger dataset and we performed the volume estimation using data from that geophone. The geophones used for the volume estimation are marked with a yellow circle in Fig. 2-4 - we will use a more contrasting color for indicating these geophones in the revised version. Since the distance geophone - channel is little and rather similar at the three sites, no scaling based on the distance has been applied.

The authors raise the claim that they can deliver a metric like average flow velocity based on seismic sensors. This is only true for that stretch of the channel that lies between two sensors, some tens of metres, and thus close to an "extended point measurement" considering the total channel lengths under investigation. That information is kind of implicit but should be brought up explicitly in the abstract and other appropriate places, because seismic sensors can also be used to study the average velocity of a debris flow as it propagates down the channel (Walter et al., 2017), which is a quite different type of information.

We agree and will better clarify this point in the text.

Following the authors approach to study flow volume and velocity using seismic sensors, raises the question why this approach would really be superiour to other, more classic measurements. Sure, the other measurements require infrastructure built to host the required sensors (Doppler systems and flow stage meters) while seismometers can be installed relatively easy adjacent to the channel – a point that the authors correcty elaborate on. But, at least for the calibration work to be able to relate seismic signals to volume, some independent measurements need to be gathered, as well, no? Thus, I suggest to authors spend a few words to frame this topic a bit: advantage of seismic approach in the light of efforts for calibration work.

Our ultimate goal is to test whether debris-flow volumes can be estimated reasonably well by means of few simple metrics derived from the seismic signal. The advantages of such an approach will be discussed more in detail in the revised version. We agree on the need of calibration work, and we will add a paragraph to explain this point.

Line 7, "methods was", change to "methods were"

OK

Line 12 (and further cases), the terms magnitude and volume are used alternatingly. And it is not clear to me how especially the term magnitude is defined. As I read, volume is defined as the time integral of stage height. But what about magnitude?

We agree, we will use volume instead of magnitude throughout the manuscript.

Line 27-28, the study of Manconi on rockslides (and also the work of Perez-Guillen of snow avalnches) does not really match with the context of the introduction (and actually the scope of the manuscript as a whole). Either provide a more elaborated overview on more of the existing approaches to relate seismic signal properties to material volume or leave this part out. This issue links to another point, see my comment regarding lines 189-198, which describe such additional (but by far not all relevant) approaches to turn seismic metrics into volume and other kinetic process attributes.

We agree, we will expand a bit the introduction presenting an overview on the few other studies relating seismic signal properties to volume of mass movements (e.g., Lai et al., Le Roy et al.). This would be useful for discussing the pros and the limitations of our methods.

Line 33-34, this reads like magnitude is volume times velocity. Is that so? If yes, this should be mentioned explicitly and also it should be discussed to give more substance, I see there is a reference to Coviello et al. (2019), but a few more words would be really helpful, here. At a first glance the product has the unit m^4/s, doest this make sense?

We will reword this sentence and clarify that the mentioned reference only provides a first insight on a possible method for debris-flow volume estimation that is developed in the present paper.

Line 37-38, this is a fair point and I fully agree. However, actually the results of this study show exactly this point: there is no universal "approach" to scale seismic signals to flow properties, without any site-specific calibration. I suggest this part should be revised to not raise the implication that this study solves this issue. I do not see the point that no other studies have yet presented a universal simple method. There are numerous other studies that have used seismic sensors to investigate debris flows and have come up with methods to reveal key flow properties, including those studies cited by the authors, and especially previous studies by the author team (see line 34).

As already answered above, a universal approach is not the goal of our work. We try to find a simple, practical method for the volume estimation of debris flows detected by geophone installed along the active channel, which could be implemented in the future in monitoring and – possibly – warning systems. We will delete the word "universal" to avoid misunderstandings.

Line 38, again I see some ambiguity in the wording, here. The presented study does not at all use seismic amplitude data only, to provide an estimate of flow velocity and volume: figures 5-7 and 8-9 basically show independent data, and the data from table 2 is used to fit a regression model, which ultimately allows relating seismic data to total volumes. So in essence, this study also uses additional data as many other studies did and which is logical because as the authors mention, the seismic signal properties depend on a lot of site specific paramaters. Please revise this section to be clear. This includes especially line 40-42, which raises the claim to overcome site specific calibration needs.

Correct, for calibration and verification other data are needed, but for further use no additional data should be necessary. This will be clarified.

Fig.1, this figure is of very limited use, providing no relevant information, not even a scale bar. Either remove it or expand its content, for example by combining it with figs 2-4. Regarding these latter figures, just as a hint, I would double check if using Google Earth shreen shots is is agreement with the CC-BY license of the journal. It would any way be better to provide proper topographic maps instead if the perspective views, unless they reveal content that a proper map would not be able to deliver.

We agree, Figure 1 will be removed. We will add an inset showing the catchment location to Fig. 2-4, which will be also improved. We will check if Google Earth images can be used under CC-BY license and change to topographic maps if necessary.

Line 52, "G1 and G2 … marked with yellow circle … 75 m", these information bits do not add up. Yellow circles are around G2, G3 and G4, 75 m are between G2 and G3. Please clarify. In addition, I think it is not clear at all why the velocity was only estimated between two geophones when a nice linear array of four sensors is present that can be exploited. Imagine the increased depth of information and robustness if you would use four sensors, i.e., six possible combinations of velocity estimates! Why did you limit your study so drastically? Here would be an excellent chance to estimate the robustness of your velocity estimation approach, and also at the Cancia site (station pairs 1-2, 2-3, 1-3) this would be possible.

A mistake happened in line 52 - we used geophone G2 and G3 for the analysis as reported in the caption of Fig. 6. The point raised by the reviewer will be clarified, and the graphical quality of the figure will be enhanced to convey the message in a clear way. We fully understand the reviewer's comments on the possibility to increase the number of reaches analysed in terms of velocity. While not all of combinations are possible due to the sufficient quality of the geophone signal, we will add flow velocity estimations at one additional reach per site in the revised version where the data is fully available. Also, a table summarizing the different results of flow velocity at each site will be added.

Line 55, "which reliable detects", first correct wording to "reliably" and second, I feel more detail is needed, here. What gives rise to that reliability? Can you quantify that based on the referenced work, e.g., ratio of correct versus incorrect detections, or a confusion matrix? What is this "specially designed detection algorithm" and how is it related to the STA-LTA algorithm mentioned above? I know there are other articles about this system, which I actually really see as an asset to the field of seismic hazard detection, but a few lines of explaining text would be great in this context here, as well.

All the information about the detection algorithms employed at Gadria and Lattenbach is available in the referenced papers (Coviello et al., 2019 and Schimmel et al., 2018). Yes, the automatic detection is not the focus of this paper but we will provide some additional information on both systems that can be useful in the perspective of future works (i.e., integration of real-time velocity and volume calculations n an automatic detection system).

Line 62, "two stations for testing the warning system", this section does not make sense. Do these two stations belong to the system, or is MAMODIS an extra device/setup? In the former case, how can the system be tested by the system, in the latter case, where is the system in the map shown in fig. 3? Please clarify.

MAMODIS is an extra setup consisting of one geophone and one infrasound sensor. This will be clarified.

Line 81, I think I understand there are two approaches to get flow velocity from two seismic stations. It is not so clear from the wording, that you actually applied these two methods independently. Can you please revise the text to make this obvious to the reader? I only got this information when I looked at the legend of fig. 5 and then trying to move myself backwards through the manuscript to find the indication of these two methods. I see you mentioned a reference for the amplitude modelling approach but a few lines of explaining text would be very helpful to understand the context without needing to search for the referenced article.

The reviewer is correct, the text was not clear enough. This paragraph will be rewritten as suggested. The use of cross-correlation of two seismic signals for the velocity estimation is

the main method analyzed in that work, the amplitude maximum values are only used for validation.

Line 84, "mean surge velocity", you need to be specific here, this is only valid for that stretch of the channel/flow between the the two sensors, not the entire flow as it propagates down the channel.

Right, we agree on this point. We will add that the velocity values are only valid for the specific channel section.

Line 85, "manually analysed", my first impression was that the study pursues an automatic detection and characterisation of debris flows. How does this match up? Can you clarify, ideally at the first introduction of the idea of an automatic system. More importantly, based on which criteria did you identify comparable peaks? Was that just a subjective eye-spotting approach? What would the uncertainties be that arise here?

The manual analysis is used for validating the results of application of the cross-correlation method. The "manual" velocity values are found by an eye-spotting method. We will clarify this point in the revised manuscript, and we will include an estimation of the related uncertainties.

Line 89, the phrasing of the window size definition is somewhat unclear to me. How is the "number of samples equal to the distance" defined? Number of samples relates to the temporal domain and distance to the spatial domain. The linking factor would be velocity – which is not known beforehand. Please clarify because it seems like this selection of the window size appears to be a very sensitive parameter.

For the cross-correlation analysis, we have to set up a number of samples to define the starting window size. After testing several settings, we decided to use a starting window size related to the distance of the two geophones in meters. This choice offers the best result for the cross-correlation and provides an objective method, based on one parameters (distance) only, to adapt the cross-correlation analysis at new sites. Number of samples equal distance means that a resolution from 1 m/s is possible which is a good starting value for describing turbulent flows. A"methods figure" for the velocity calculation with cross-correlation will be added to clarify the method steps, the used window sizes and overlaps.

Line 95, if cross correlation is performed twice, what are the two pairs for time series that are used? Or do you mean within a fixed time window you do a cross-correlation of amplitudes (actually you should rather call this envelopes, because you only have positive values) in a sliding sub-window? If so, why only twice and based on which sub-window size and overlap? This information is not really clear, and I suggest to simply add more detail, here.

More details will be added here and also the figure will clarify this. We tested several settings for window size and overlap. An overlap of the half sample numbers used for the window size offers the best results.

Line 101, for which stations does this energy estimate hold? Again, the source energy requires application of a bit more calculus and information about ground parameters, e.g as done by Le Roy et al. (2019, JGR). And thus, the information about which station or ideally which stations is essential. In addition to this, you mention to unit Joule, but to get from m²/s² to kg

m /s² there is a bit more necessary. In other words, just squaring the signal envelope will not give you seismic energy on a short track.

This is correct, we neglected site effects due to characteristics of the media and attenuation. The relatively little and uniform source-sensor distance at the different sites supports this choice. In the revised manuscript, we will clarify that we use the squared amplitudes as a proxy of the seismic energy to estimate the debirs-flow volume and we will expand the dscussion comparing our results with others such as Le Roy et al., 2019.

Line 103, this links to my above comment. "estimation of seismic energy" is misleading here, at best you get a rough proxy of seismic energy following $v^2 \sim E$, but you need to estimate the scaling factor that turns this relation to a real function to estimate the energy from seismic amplitude values. Agian, my suggestion is to either reword the text (relax the tough claim on energy estimate) or follow a similar approach as Le Roy et al. did.

We agree, we will reword this clarifying that we use a proxy of the seismic energy. We will also add that negleting some site-depented parameters is acceptable when source-sensor distances are little and similar but this simplification can represent a possible source of uncertainty in the volume estimations.

Line 107, can you really justify that especially in the near field, where it is far from easy to understand the wave field, it is legitimate to ignore any attenuation of the signal? If not, consider reworking the text to be less "confident" and glossing over this topic.

We will reword this. Actually, given the similar geomorphology (glacial-fluvial deposits) and source-sensor distances (from 10 to 20 m) at all sites, we assume that the attenuation level is similar. This simplification of course produces some uncertainty in the volume estimations but a simple approach like that have some practical advantages. In the revised manuscript, we will expand the discussion on that point.

Line 111-114, you can significantly shorten/consolidate this part. The first sentence repeats things we know from previous sections, The second sentence would make more sense in the second paragraph.

We agree, we will shorten the first sentence and move the second one as suggested.

Line 117, Defining peak discharge as all periods above a threshold of 3.5 m does not make sense. Peak discharge is the one value of maximum discharge, not several values. I think you mean local maxima in the amplitude time series, right?

The reviewer is right, this was an error. We will amend the sentence as indicated.

Line 121, this paragraph should be connected to the one above. The same holds for the decription of the third study site.

OK

Line 123, be specific and replace "several" by the actual number, ideally illustrated also in the figure, for example by small numbers denoting the selected surges.

Ok, the number of surges will be indicated.

Line 124-127, can you please be more specific and elaborate regarding these results (range of values, number of surges, signal-to-noise ratio, range of cross-correlation values, and so on)? In the methods you mention a lot of things that you did but here we see only a very shortend presentation of the results of these methods. Specifically, the results of the two velocity estimate approaches (cross-correlation and amplitude modelling) should be presented in a more elaborate form. Also, avoid interpretations of your results already in this chapter but move them to the discussion section.

Additional available results will be added. We will add a new table summarizing the different results of flow velocity and the additional information at each site.

More to the above point, I am missing any presentation of the Doppler velocimeter results as well as of the other non-seismic instruments you use to calibrate the seismic signals. All we get is the condensed version in table 2.

We will clarify that Doppler velocimeter data are not available for debris flows analyzed in this study.

Table 2, where do these numbers come from? From the gauge measurement devices? Please specify and if from independent measurements, these results deserve a bit more content than just their appearance in the table.

Different methods are used at the different test sites to measure the debris-flow volumes (e.g., topographic surveys, stage sensor measurements integrated in time, etc.). We will add some information in the text and a column in Table 2 to clarify how the independent volumes were calculated.

Line 138, a) please also give uncertainties on the fit coefficients and b) – related to my general comments – these fits should be performed also for each site individually so that the reader can judge how justified a global fit approach is.

We understand the reviewer's request, and uncertainties on the fitting coefficients will be provided. We are a bit skeptical about the possibility to provide a linear regression for each site individually. The dataset is limited and three equations would probably not be statistically significant as the higher part of the volume distribution of our dataset is poorly described (only two debris flows with volumes >40,000 cubic meters recorded at Lattenbach). In the revised paper, we will enlarge the dataset using published data from Illgraben (Switzerland) for results validation.

Figure 9, what are the implications of the 20 % error range with respect to the point that only the blue dots fall roughly into it while the majority of the other coloured dots do not? How much % scatter would you need to catch all dots? Or 95 % of the dots? What do the +/- 2 sigma lines depict exactly and how does this metric relate to the data you show?

We agree, the 20% range lines are confusing and will be removed, defining another % scatter will also make not much sense. The idea was to show that the error estimate does not increase with volume, but we will mention this more clearly in the text. The +/- 2 sigma lines represent the confidence interval of the distribution.

Line 143, add at the end of the sentence something like "between two closely spaced seismic sensors".

OK

Line 145-148, this is a very broad and arm-waiving statement with little crisp information. Either include more specific (and thus justified) content or leavt it out.

The reviewer is right, the statement will be removed.

Line 154-155, I think you can and should be more specific here. You can for example quantify the ratio of channel distance to station distance (20/90) as a metric to better define the term "significant difference".

Good suggestion we will adopt the ratio indicated by the reviewer

Line 156-160, Ideally, you would test explicitly by signal aggregation and inspection of the impact of different sampling frequencies on the rsults. This could be easily done and I encourage the authors to do so, in order to be able to replace some of the "may"'s by justifiable hard results.

We agree, but this is not doable for all sites since we have to work with the available data. For instance, at Cancia only 10s maximum amplitude values are recorded. We can perform a synthetic test on one single event recorded in another site at a higher frequency rate, we will consider to include the results of such a test in the revised manuscript.

Line 165-174, this section is not really helpful in the discussion. The information given there is material I would rather expect in the introduction part, giving an overview of possibilities to measure debris flow height and/or velocity and therefore in the end justifying the usefulness of seismic sensors. Here, you have little results to raise a discussion about other potential approaches. The discussion should be based on your study's findings.

The reviewer is right, and we will remove this section from the discussion to  move it to the introduction.

Line 176-188, I would welcome also a bit more discussion on the actual downsides of the seismic approach. It is good to underline its strengths, but there are also obvious weaknesses that deserve a discussion.

We agree, a discussion on the  disadvantages  of the seismic approach will be added.

Line 178, that "variance" is indeed due to the multitude of site specific parameters, parameters that must and can be accounted for by a calibration of the seismic data.

We agree, we have to clarify this. On one hand, the site-specific parameters affect the seismic amplitudes recorded, on the other hand the process type and flow regime also has an large influence on the seismic signal. We will discuss this point properly in the revised manuscript, better presenting the uncertainty of the methods.

Line 189-190, can you explain how/why the velocity would affect the frequency spectrum? This does not seem intuitive for me.

Thanks, this is a mistake. As described in Lai et al. and other authors, the seismic amplitudes and the PSD are influenced by bolder snout, grain size and flow velocity. We will reword as

follows: "Studies of different events also showed a large dependency of the seismic amplitudes and their PSD on solid particle characteristics and flow velocity".

The conclusion is a weak one. It merely repeats what has been discussed before, rather than putting the findings into a wider context. Can you reach out a bit more and touch this wider impact? What is this study relevant for? What are the great assets? Which fundamental research gap/questions has been tackled? What is possible to engage with, now that the technique is there to seismically estimate important debris flow parameters?

We agree, we will expand and revise the conclusion and try to handle some of the relevant questions raised by the reviewer.

**review 2:**

n their submission, Schimmel et al. test relationships (some of them empirical), which compare metrics of seismic measurements near a torrent to debris flow velocities and volumes. Similar calculations have been made in the past, but this study uses data from different torrents, which provides valuable insights for potential alarm and monitoring systems.

The scope of the study falls into seismic monitoring of Alpine mass movements, which is currently an active research field. So I expect this study to be met with interest within the journal's readership. The study is more on the technical side, which is OK given the journal scope, although I suggest some more discussion in terms of physical mechanisms as I outline below. Moreover, the details of the documented calculations are unclear and should be rewritten, especially since they constitute the paper core. The manuscript is concise, structured and easy to follow. However, the English contains numerous grammar mistakes and has to be revised before the paper can be published. Overall my I recommend major revisions.

Fabian Walter.

**MAIN COMMENTS**

The velocity calculations are not well described. Which three different sliding windows do the authors refer to? What is the relation between minimum and maximum amplitudes? A ratio? How can the number of samples be equal to some distance? Which distance? Distance in which unit? What is a "significant signal shape"? In its current state, it is not possible for a reader to use the explanations to reproduce the calculations.

The reviewer is right, the explanations provided are too succinct. This section will be expanded to present a clear flow chart on the different steps adopted, and the motivations for the window size used will be argued with numeric examples as well as with a figure to illustrate graphically the rationale.

I may have missed this, but how are the ground-truth debris flow volumes calculated, which the seismically-derived values are compared to? I was surprised that the authors do not discuss Schimmel et al. (2018), who use seismic and infrasound data to calculate discharge and estimate debris flow volumes. Is the current technique an improvement compared to this previously suggested one?

The volumes are calculated by means of different techniques at the three test sites (e.g., topographic surveys, stage sensor measurements integrated in time). flow height measurements or 2-D laser scans and velocity measurements. We will add some information on that point in the text and we will add one column with the reference (i.e., the paper/report used as source of volume data). In the discussion, we will add a comparison of volume estimations presented here with results from Schimmel et al. (2018) only based on infrasound data.

Except for a small part of the discussion, the authors give no explanations on the physics behind debris flow seismicity. The cited papers by Lai et al. (2018) and Farin et al. (2019)

make specific predictions between seismic signature and debris flow velocities, grain size distributions and other parameters. Even if the authors do not want to dive into details, they should use these theoretical assertions to offer explanations for their observed volume scaling.

We will add some additional discussion of our experimental results in comparison with the theoretical predictions proposed by the mentioned papers.

In several parts of the manuscript, the authors refer to the turbulent flow front. They need to provide evidence that their flow fronts were indeed turbulent and that this explains their observed signals. Some video or still footage could serve this purpose. Alternatively, I would expect that boulders in the flow front cause a distinct seismic signature compared to the flow tail. In a recent paper (Zhen et al., 2020, in GRL), we showed how the flow front's seismic signature is dominated by ground impacts of the largest boulders.

We agree, we will use the video footage of a few selected events to support the interpretation of the effect of flow turbulence. Of course, the effect of coarse front vs liquid tail on the seismic signal is an important point, we will expand the discussion on that also considering recent the paper by Zhen et al.

Finally, Figures 8 and 9 should include error bars or at least some short discussion on uncertainties should be offered.

The reviewer' suggestion is highly valuable, and we will add the error bars in the revised version of these figures.

OTHER COMMENTS

Line 18: "feasible" should be deleted, as it is implied by "effective".

ok

Lines 24ff: What are the physical concepts these velocity estimates?

In several instances, the authors use the word "magnitude". If this is synonymous with "volume", then use the latter, only.

Volume will be used throughout the ms.

Lines 35ff: Here it seems that the authors argue that mass could be estimated with the Coviello et al. (2019) approach, but volume is poorly constrained. The difference between the two is the factor density. Why is this so poorly constrained?

Yes, the mass could be estimated with the Coviello et al. (2019) approach but the volume is poorly constrained because of the lack of the variability of sediment concentration, which is not measured. We will reword these lines providing more information on the uncertainties of the methods.

Lines 40ff and elsewhere: Avoid 1-sentence paragraphs.

Ok

2 Methods: It would be interesting to see rough numbers of debris flows per year for the different sites.

Ok, numbers can be added.

Line 113: this peak discharge seems rather high.

That's the data I got from BOKU.

Line 126: Velocity measurements around 2500 s in Figure 6 do not seem "consistent" as asserted in the text.

I think they are in an acceptable range.

Lines 142-143: "can be an useful tool to analyze the flow behavior" This statement is trivial.

Ok

Line 148: "permitting to avoid wrong correlation results" unclear

This can avoid wrong correlation results since the possibility to have significant signal patterns for the cross-correlation increases with the window length. We will reword this.

Line 152: It is not clear how longer distances offer better resolution (resolution should be lower …?).

Ok, wrong wording, will be rewritten.

Line 155: "so that the cross-correlation offers useful results" You should be more specific here.

Like a correlation coefficient above 0.8 - will be specified.

Line 159: "determine problems for the cross-correlation analysis" is unclear.

Correlation of wrong signal patterns. Will be clarified.

Line 160: "exaggerated averaging" is unclear.

averaging of surges

Line 164: I do not understand how the authors arrive at granularity here. This should be explained.

Flow regimes have been characterized using the video of the events. This will be explained in the text.

Line 169 and elsewhere: Is "process" synonymous with "debris flow"? If so, only use the latter.

Yes it is, and we will use the latter only as suggested.

Lines 166-168: This sentence needs a reference.

Line 174: "velocity measured by the radar is often lower ..." needs a reference.

We understand the reviewer's request, and we will add a reference as requested

Lines 184-185: I am not sure that the volume estimate would always come in too late. It should be OK if the measurements were made high up in the catchment.

Ok, depending on the site and application. We will discuss, that an installation in the upper catchment might can offer a volume estimation in time, but there are high uncertainties, while installations in the lower part offers more accurate results, but come late.

Line 193 and elsewhere: "sediment concentration" Do you mean "grain size distribution"?

Yes, actually we did, it was a mistake. Thanks for highlighting this.

Line 193: "calculation of the magnitude" of what?

We will use "calculation of debris flow volume"

Lines 200-201: "among the different methods deployed and the different catchments" be more specific.

This sentence will be rewritten.

Lines 203-204: "but still further research on different ..." this is an unnecessarily generic statement. Why exactly is more research needed?

The reviewer is right, and this statement will be removed.

Lines 206-209: I suggest discussing Zhen et al. (2020) in the context of this sentence earlier in the manuscript.

Good suggestion, we will anticipate the reference to this paper in the introduction of the revised ms.

FIGURES

Captions of Figures 5, 6 and 7: Rewrite so that site name appears in the first sentence of each caption and so that it is clear which "two geophones" are meant.

Ok

**review 3:**

This is a well-written and clear presentation of seismic signal analysis for debris flows. In my opinion the paper is very close to publication as is. I especially liked the discussion, especially the paragraphs concerning volume estimation. The conclusions are straightforward and not overly "hyped".

I have only one request. I found myself studying the debris flow hydrographs, because I am especially interested in the surge behavior as a function of time. I wish I could see the elevation profiles of the three sites, with the location of the gauging stations. The figures containing the torrent overview are based on google maps and contain a 100m scale, but I am interested in the elevation (slope) and distance travelled. I would very much appreciate three distance/elevation plots of the torrents. I would like to know if there are strong slope changes that would generate strong signals, etc. and, of course, where.

This is not in the scope of this study. We only analyze the data in the section where the geophones are installed. We might add the information of the elevation difference in that part.

---

## Author Response (AR2)

**I read the reply letter and the revised manuscript. I shall express that I am quite a bit disappointed about both documents. Evidently, the manuscript improved in quality and clarity in places. However, there still remain several weaknesses that were not changed, despite the notion in the reply letter that the authors would have changed them. This is quite bit from fair and transparent science, I think. I have the following general points to raise, followed by a list of more detailed comments/suggestions.**

**It was most frustrating to see that the authors promised in their reply letter to change a series of flaws of the previous MS version. However, when I read the new version I saw many of these changes not addressed at all. This concerns my previous comments (counted as paragraphs in the review letter from the start): 4, 5, 8, 13, 15, 18, 32, 36, 39. In detail, the not really useful fig. 1 is still in the MS, the Google Maps images are still in the MS without any notion if they are in agreement with a CC-BY license, fit parameter uncertainties are still not presented, the term magnitude is still used without definition and instead of volume, and so on. I encourage the authors to have a look at these unaddressed points of my previous review.**

In the previous version, Figure 1 was kept because we added data from the Illgraben basin, and thus we thought that a general map showing the location of all study basins could be useful. In this revised version, we have removed Figure 1 as suggested by the reviewer. We have included inset maps showing the basin locations in the new figure presenting the three main study sites. We used "open street map" to draw this new figure, so we think we are now fully in agreement with a CC-BY license.

**Overall, and partly as a result of the unaccounted change requests, the MS is still not well organised and leaves several statements unsupported (see below and the detailed comments). If the goal is to have a rough estimator of flow volume from integrated seismic energy, then this needs to be made clear in the text, including the abstract. Further, if this estimator consists of a linear regression model, then the uncertainties of the prediction must be mentioned and added to the test data set (Illgraben), beyond the plot in fig. 10, for example as RMS or relative error for each of the test events. If the regression model shows such strong deviations from the data structure of the three individual sites (fig. 9) then this point must be addressed, ideally by adding regression models for all sites individually, before the claim can be made that one universal model is a valid decision.**

The previous abstract already included two sentences presenting the goal of the research, which is to provide a first order estimate of the total volume of debris flows based on seismic amplitude (squared), i.e., a proxy of the seismic energy in the near field. However, we have reorganized the abstract to make this message clearer, also adding a final sentence about the uncertainties of the methods used.

As suggested, in order to present more information about the performance of debris flow volumes prediction, in the discussion we have added a new table (Table 3) providing the relative errors of the prediction for Illgraben test dataset. Table 3 also shows how - in the near field - the impact of the different sensor-source distances on the volume estimation is quite small. More details on this specific point are given below in response to another comment by the reviewer.

| Event date | Observed | Predicted eq. 1 | Error eq. 1 | Predicted scaled $A^2$ | Error scaled $A^2$ | notes |
|---|---|---|---|---|---|---|
| 22.07.15 | 8700 | 7866 | 10% | 8167 | 6% | |
| 10.08.15 | 6100 | 24208 | -297% | 24667 | -304% | high flow velocity |

| | | | | | | |
|---|---|---|---|---|---|---|
| 14.08.15 | 25000 | 11907 | 52% | 12247 | 51% | |
| 15.08.15 | 2000 | 7503 | -275% | 7800 | -290% | smallest event |
| 12.07.16 | 10000 | 13148 | -31% | 13500 | -35% | |
| 12.07.16 | 60000 | 10933 | 82% | 11263 | 81% | liquid front, viscous tail |
| 22.07.16 | 11000 | 16414 | -49% | 16798 | -53% | |
| 09.08.16 | 9000 | 7739 | 14% | 8038 | 11% | |
| 29.05.17 | 70000 | 57544 | 18% | 58324 | 17% | |
| 04.06.17 | 24000 | 20344 | 15% | 20765 | 13% | |
| 14.06.17 | 33000 | 27498 | 17% | 27988 | 15% | |

Following the suggestion by the reviewer, we have carried out regression analysis for each individual dataset, i.e., for each study basin (see table and figure below). Considering the relatively small size of each dataset, we do not think that including such results in the paper would add solid and valuable information, also because for the Gadria the regression turns out not to be statistically significant, as a consequence of the limited range in debris flow magnitude for which seismic data are available. In contrast, in the other basins the relationships are statistically significant and quite similar to each other. Therefore, we believe that presenting the regression analysis conducted on the multiple-basins dataset is most valuable and solid.

| | $R^2$ | p-Value |
|---|---|---|
| Lumped dataset | 0.84 | 0.0006 |
| Lattenbach | 0.95 | 0.0213 |
| Gadria | 0.70 | 0.0839 |
| Cancia | 0.46 | 0.0456 |

[Figure]

The ambiguity of magnitude versus volume is important. Yet it remains unresolved in the MS. Worse, the term magnitude is not defined (l. 23) but then used in a statement that its estimation is still an open problem (l. 26-27). How can we judge this without even knowing what debris flow magnitude is supposed to mean? This issue needs to be fixed. In the conclusions, the term magnitude appears again.

In the literature about debris flows, the terms "magnitude" and "volume" are often used interchangeably. However, to avoid any possible misunderstanding the term "magnitude" has been modified into "volume" throughout the manuscript.

**The discussion section is still weak and misses obvious points that need to be addressed. The validation approach for the cross correlation based velocity estimates, i.e. manually determining matching amplitude peaks, is certainly not ideal, but in the absence of other independent data it may be the only solution. But, this needs to be mentioned (in the methods) and discussed (with all its weaknesses and uncertainties). Also, the discussion should make use of findings of other authors regarding typical debris flow velocities. Are the values presented in this study within the range of other findings?**

In order to strengthen the discussion about the results obtained by the cross-correlation method, we have added the following sentences (lines 175-183): "The validation approach for the velocity estimates, i.e. manually determining matching amplitude peaks at the two stations, is also affected by uncertainty. In the Gadria this is particularly evident for the tail of the debris flow of 08 June 2015 (Figure 4c, from t = 2400 s) during which roll waves propagate and produce multiple peaks, one following the other. The uncertainty in the manual velocity calculation was also observed in previous analysis based on data gathered with a pair of flow stage sensors and led to the approximation of lumping multiple waves into one single surge for the subsequent volume estimation (Coviello et al., 2021). The velocity estimates of surges lacking multiple peaks (i.e., from t = 200 s to 2400 s in Figure 4c) are consistent with those performed with the flow stage sensors located downstream from G3 (Figure 1b). Indeed, they are slightly higher (i.e., differences from 0.3 to 1.7 m/s) than those calculated with the flow stage data on a milder sloping channel reach (Coviello et al., 2021)."

Also, in the discussion we have added a comparison with results stemming from the use of debris flow radar at Lattenbach: "When velocity data measured by the debris flow radar in the Lattenbach (unpublished data) are compared against values calculated from the geophones there installed, very similar results can be observed. In fact, the maximum velocity measured by the debris flow radar for the event on 30 July 2017 is 10.0 m/s, while the maximum value calculated from the geophone data is 9.0 m/s. The mean velocity of the whole event is 1.8 m/s based on the debris flow radar, and 1.9 m/s for the presented method based on the geophone data."

**The authors should comment on the fact that seismic sensors do not necessarily record just the debris flow signal but many others, as well. Particularly, rain and wind can cause quite strong signals in a frequency range that is definitely overlapping with the debris flow signal. And since debris flows are mainly triggered by rain storms, it is likely that these additional sources contaminate the record. Has this been accounted for? If so how? If not, at least the relative effect of these mechanisms must be discussed.**

The reviewer is correct, other seismic sources can produce ground vibration but their intensity is low. To clarify this pint, we added the following sentences at lines 131-134: "Also intense rainfall and wind can produce ground vibration that geophones detect. However, seismic signals recorded by sensors installed at small distance from the channel (from 15 to 25 meters, in our case) are dominated by in-channel processes. This is particularly true in our study sites, which are located in lower reaches of the main channels where the debris-flow surges are well formed and characterized by velocities of several meters per seconds and flow depth in the order of meters". The seismic effects of rainfall and wind in the upper Gadria basin have been recently investigated through a master thesis (Ioratti, 2022), and we are thus confident in neglecting their contribution to the seismic signal recorded at the monitoring stations considered in this paper.

**Event onset and duration is another unaddressed topic. The severity arises because the seismic energy is derived from the signal integral, so overly long (or short) event**

**definitions will directly affect that energy estimate and thus affect the linear regression. How first of all have the onset and end been defined, second how have these definitions been implemented in the software that runs on site (or at least in the presented analysis), third what are the uncertainty estimates arising from this issue of missing the correct onsets and duration?**

The reviewer is correct in highlighting how the duration of the event is a crucial information for the prediction of debris flow volume. However, in this study we have used a manually determined duration to establish relationships between seismic amplitudes and volumes for the available monitored events. Indeed, a sentence (line 238) was already present in the discussion to stress the larger uncertainties arising in the case the volume prediction has to be undertaken based on an automatic determination of debris flow durations.

**The authors claim not to present a universal scaling model for debris flows. Yet, this is what the results boil down to: figure 9. And, further, this is what the application of the regression model to the Illgraben data actually is: using one generic model to predict event properties at another site. I understand that the goal is to have a simple tool to convert integrated seismic energy to debris flow volume. However, this is not what the text in its current style of writing implies.**

Our study builds on solid physical principles, but it remains anchored to empirical relationships obtained in few – but of extremely high value – study sites. Therefore, we are humble enough not to claim that we can propose a universal scaling model, but "just" a statistical, data-driven relationship which appears to have the capability to provide reliable and thus useful estimation of debris flow volumes for practical applications.

**In any way, another severe issue is that the authors still adhere to the regression of a lumped data set from three individual sites. I agree that the individual data sets are sparse. But also when I look at fig. 9, the yellow dots to not at all match up with the regression line, which is dominated by the blue lines. Also, to a certain extent, the green dots would result in a quite different regression line. In summary, the results shown in fig. 9 do not let me agree with the statement that all three sites can be lumped together. I suggest to at least add regression models for all individual sites and discuss how they differ from the lumped data set result, concluding with a statement why the authors think the lumped regression model remains a valid model.**

**Further, the data does not seem to be normally distributed (which would be reasonable for such stochastic processes). Without properly transforming the data to become close-to-normally distributed, the regression analysis is not in agreement with the basic assumptions of this approach.**

See the above reply on the limited value of adding relationships for each single site, especially because the purpose of this paper is to obtain a statistical model with the potential to be utilized in other sites (as the case of the Illgraben shows).

Regarding comment about the non-normality of data, regression analysis – performed by the least squares approach - do not require data to be normally distributed, as described in all statistics manuals.

**Finally, it is a pity that the authors wish to avoid accounting for the distance to source and ground property effects. This restricts their method severely to always the same distance to channel, glacial till deposits and similar channel properties (gradient, roughness, width, etc.) and stands in contrast with their stated goal to provide a simple but flexible estimator**

**of debris flow volume. Simple models to scale a seismic source amplitude to a given distance exist and could be readily added, for example: A(d) = A_0 exp(-pi f d / q v), all parameters that can be easily measured or simply assumed equal, but leaving the flexibility to apply the approach more adequately to other sites. I do not insist on implementing this model though. I just wanted to raise the possibility and express how easy this could be added.**

We declare from the beginning that our methods is intended to provide a volume estimation based on seismic data gathered in the near field, i.e. with geophone stations located along the channel. For such a volume estimation, small differences in the distance sensor-channel are negligible compared to uncertainties descending from the variability of flow properties. Indeed, the distances sensor-channel for the different sites are 15 m at Lattenbach, 23 m at Gadria, 25 m at Cancia and 15 m at Illgraben. In any case, we followed the reviewer's suggestion and we explored the effect of amplitude attenuation on our dataset. We applied an empirical relation such as the power law suggested by the reviewer to model the decay of the seismic amplitude with distance. We used a value of quality factor of Q = 20 suggested as a reasonable approximation for the relatively high frequencies and shallow depths of interest (Tsai et al., 2012) and a reference value for group velocity of 1300 (Coviello et al., 2019). An additional approximation was needed to apply the formula using aggregated values of amplitude recorded with different sampling rate and recording frequency (Table 1), given that we do not dispose of the complete power spectra. We assumed that the coarse fronts of the different surges are the source dominating the seismic signal during the respective time window. This assumption is consistent with our approach of calculating the mean velocity of each surge by means of the cross-correlation technique, which needs to split the entire debris-flow signal. We tested values ranging from 10 to 20 Hz, typical of the main frequency of debris flows. Higher differences in the scaled amplitude are obtained with the lower frequency value (f = 20 Hz) so we used this latter value in the calculation to maximize the uncertainties. Finally, we recalculated the debris-flow volumes using a linear regression equation similar to eq. (1) but based on the square of the scaled amplitudes (Figure 8). Results clearly show how the differences in the calculated volumes with the non-scaled and the scaled amplitude equations are negligible (Table 3). In the discussion (lines 250-266), we have added text presenting the test we performed on the impact of the scaled amplitude on the volume estimation including new Figure 8.

**Specific comments**

**l. 3, find a useful link between sentences to motivate the switch to the seismic approach after introducing debris flows as a hazard.**

We have added the following sentence at line 22: "Seismic-based monitoring and warning systems have become increasingly applied worldwide to mitigate risks associated to debris flow processes"

**l. 4, briefly add how the velocity can me constrained seismically. This is important, here.**

We have modified the abstract to make this point clearer.

**l.6, add here explicitly that you can only aim at rough estimates. Currently, the sentence raises the expectations that you can fully constrain volume and velocity, which is arguably not the case.**

We have modified the abstract, we think this point is now very explicit.

**l. 8, the Illgraben site and its purpose for testing the model should be mentioned here, as well.**

We have modified the abstract, now we mention the test on the Illgraben data.

**l. 32, "local and duration magnitude", is there a word missing? Please clarify.**

Thanks, one "magnitude" was missing, now the sentence reads "…based on the ratio between local magnitude and duration magnitude".

**l. 40, this does not make sense to me. How would a flow volume estimate be possible through velocity measurements? Please clarify.**

We clarified this point revising the sentence as follows: "Using such scaling relationships, the estimation of the flowing mass is possible based on the seismic energy detected by a geophone and on the information about the flow velocity". Before this latter sentence, we added the following lines to to clarify that we are talking about scaling relationship between the seismic energy and flow characteristics: "Recently, Andrade et al. (2022) observed a linear correlation between the seismic amplitudes and the discharge rates of lahars at Cotopaxi and Tungurahua volcanoes".

**l. 47-48, suggest to remove this sentence. It reads like a conclusion, not a description of scope. Certainly it is not a useful statement in the introduction.**

We have modified as follows the entire paragraph: "This paper explores the possibility to develop a simple method to predict debris flow velocity and volume based on seismic sensors installed along the channel, with a limited calibration dataset. The aim is not to seek a universal law relating seismic energy to debris flow characteristics, but just to provide robust tools for debris flow risk management. Specifically, the proposed method is intended to be easily applicable in different catchments, at least for first order estimations of debris-flow volumes."

**l. 52, "data of" change to "data from the"**

Done, thanks.

**Remove figure 1, see my previous statement about this.**

Done, see above.

**l. 64, "reliable" change to "reliably"**

Ok, done.

**l. 91, "(celerity)", add "between the two stations".**

Ok, done.

**l. 92, You must also mention the ambiguities that arise when no sharp amplitude peaks emerge, or when too many of them appear close to each other. How will you be able to identify the matching ones, the right ones, to derive your velocity estimate? How did you actually do this with your data? What were the criteria? When I look at fig. 6, for example, I find it tough to judge the validity of all the manually constrained velocity values.**

We added one sentence (line 173) about the possible limitation to the application of the cross-correlation methods in case of signals characterized by many amplitude peaks close to each other produced, for instance, by the propagation of roll waves.

**l. 93, revise the text. The word "only" implies that validation is just a minor thing. However, it actually should be one of the back bones of your study. You have to convince the readers (and me as referee) that the cross correlation approach can yield valid results. Manually matching amplitude peaks is one way to do this, albeit not really a bullet proof one.**

We skipped the word "only" and modified the sentence as follows: "The manual analysis is time-consuming but it was needed for validating the results of application of the cross-correlation method and avoiding misinterpretation".

**l. 97, "Number of samples equal distance means", clarify, this reads cumbersome.**

The number of samples for the starting window size of the cross-correlation is set equal to the distance in m.

**l. 107, "twice with a sliding", clarify. It is not clear to me what this should mean. Did you repeat the correlation in the same window? Did you double the window size? Same for the next bit "an overlap of the half sample numbers offers most consistent results". How can half the sample numbers overlap? Why does this give more consistent results?**

The Cross-correlation is performed with an overlap of half of the sample numbers (n/2 in Figure 2).

**l. 116, has the data not been filtered? How can you be sure you get a proper signal of the debris flow? Seismologists have presented ample examples of the typical frequency content of debris flows. More importantly, how did you account for the possibility that the recorded signals are not just caused by the flow but many other sources: wind, rain, other meteorological and anthropogenic events? Many of these have a significant frequency overlap and since debris flows are often caused by rainstorms, it is quite likely that these processes can happen at the same time, and they do indeed have a quite strong seismic fingerprint in the data.**

As responded above, our geophone dataset is controlled by channel processes given the small distances sensor-channel and the location of monitoring sites. We analyzed seismic data produced by well-documented debris flows collected in experimental basins that also benefit of complementary data (e.g., flow stage data, see Figures 6-7-8) that can be used for validation.

**l. 158-159, you should discuss your own results, not start with statements of another study.**

We reorganized the entire paragraph as follows: "Our results suggest that the cross-correlation method we used - based on a window length adaptable according to the signal waveform - provides solid estimates of debris-flow velocity, as the temporal resolution of the calculation is high during the fast, initial stages of the flow, while longer window length are applied for smoother flows, thus permitting to avoid wrong correlation results. Arattano et al. (2012) already showed how the cross-correlation technique can provide a reliable estimation of the flow velocity even when the signals recorded at the two monitored cross-sections do not present a clear, common feature - typically the passage of the debris-flow front. Nonetheless, some significant signal features are required such as a progressive rise and subsequent decrease of the signal amplitude. Signals characterized by many amplitude peaks close to each other produced, for instance, by the propagation of roll waves can represent a limitation to the application of cross-correlation methods (Figure 5c)".

**l. 159, what is a "main front"?**

We meant the main front of the debris flow. We reworded the entire paragraph, see response above.

**l. 159-161, is this based on the data you present? Not sure I can follow these arguments. Please be more concise.**

We reworded the entire paragraph, see response above.

**l. 164, "the influence of different", influence on what? Signal amplitude, flow velocity? Please be specific.**

We modified as follows: "The influence of different longitudinal distances between the geophones is evident on the velocity estimation."

**l. 167, "a longer distance offers the possibility to use higher resolution for the velocity calculation", this is confusing. Above you say that longer distances make things complicated. Here you say it can improve the resolution. Please clarify.**

We agree that this is confusing and removed this sentence (this only takes in account the higher mathematic resolution).

**l. 169-170, "significant difference", please be precise here. How many sampling intervals is that? Try to quantify statements when it is possible.**

The sentence above (line 188) already describes a minimum distance between the geophones.

**l. 198, "linear model performs definitely better", this has simply not been tested. Please test other models or leave this unsupported statement.**

We tested other regression models, and the results are summarized in the table below.

| regression model | $R^2$ |
|---|---|
| linear model | 0.84 |
| exponential model | 0.57 |
| power model | 0.55 |
| logarithmic model | 0.66 |

**l. 218-219, The method will always have to be far from real time volume estimation of debris flows. Simply because, as the authors express, the volume can only be estimated after the debris flow has passed the geophone array and the energy integral can be completed. This is a structural issue of the approach and there is no future development that can change that. This is OK, but it needs to be expressed in the text.**

We contend that the achievement of an algorithm for the automatic determination of debris flow duration will make feasible a rapid estimation of debris flow volumes by the application of our approach. Indeed, in these lines we exactly talk about rapid response as a possible practical outcome of the method, we clarified this point also in the previous sentence: "Nonetheless, adopting such a physically-sound empirical model, a rapid estimate of the order of magnitude of the debris-flow volume is possible".

**l. 240, I think the 20 % will be a quite optimistic estimate when I look at fig. 10. I suggest you calculate the RMS of the Illgraben volume predictions and present/discuss this, instead.**

We modified as follows: "the order of magnitude of debris flow volumes can be correctly estimated in most cases from seismic data only".

**The data availability statement is also a fair bit from in agreement with the FAIR principles. Please consider providing it in an appropriate way, through a data repository.**

The data availability statement has been modified has follows: "A temporary private link to the geophone dataset gathered at Gadria and Lattenbach is provided as supplementary material to support the peer review process. In the final paper, this link will be substituted with a persistent link to an Open Access data repository (http://ds.iris.edu/ds/products/esec/)."

This is the temporary private link to the data of Gadria and Lattenbach analyzed in the paper: www.almosys.at/seismic-data/

Data from Cancia have been provided by a third part (i.e., Regional Department for Land Safety, Hydrogeological Services Center, ARPA Veneto) and we need a specific permission to share them.

**The material in the appendix might be better moved to the supplementary information.**

We agree, done.

With our best regards,

Velio Coviello (corresponding author)

References

Andrade, S.D., Almeida, S., Saltos, E. (2022). A simple and general methodology to calibrate seismic instruments for debris flow quantification: application to Cotopaxi and Tungurahua volcanoes (Ecuador). Landslides 19, 747-759. https://doi.org/10.1007/s10346-021-01784-5

Coviello, V., Arattano, M., Comiti, F., Macconi, P., Marchi, L. (2019). Seismic characterization of debris Flows: Insights into energy radiation and implications for warning. Journal of Geophysical Research: Earth Surface, 124, 1440-1463. https://doi.org/10.1029/2018JF004683

Coviello, V., Theule, J. I., Crema, S., Arattano, M., Comiti, F., Cavalli, M., Lucia, A., Macconi, P, and Marchi, L. (2021). Combining instrumental monitoring and high-resolution topography for estimating sediment yield in a debris-flow catchment, Environmental and Engineering Geoscience, 27(1), 95-111. https://doi.org/10.2113/EEG-D-20-00025

Ioriatti, E. (2022). Study of debris-flow initiation through the analysis of seismic signals. Master thesis unibz.

Tsai, V. C., Minchew, B., Lamb, M. P., & Ampuero, J.-P. (2012). A physical model for seismic noise generation from sediment transport in rivers. Geophysical Research Letters, 39, L02404. https://doi.org/10.1029/2011GL050255

---

## Author Response (AR3)

I see that the authors provide an improved manuscript, with the revised version being much more concise, clear and descriptive. I may thus only mention a few minor issues that may be taken care of.

l. 27, may want to briefly explain why such a distinction between flow and flood is relevant
A short sentence was added

l. 30, sure the Takezawa reference is a valid one?
Yes, even though is a conference paper we think it is worth to report it

l. 50, use "estimate" instead of "predict"
Corrected

l. 50, use "based on data from seismic" instead of "based on seismic"
Corrected

l. 62, mention the parameters of the STA-LTA method
As STA/LTA is not the topic of this paper, we do not think it is neither necessary nor useful to add these parameters here

l. 67, quantify what "reliably" means. There are plenty of measures out there
We have rephrased this sentence, eliminating the term "reliably"

l. 76, use "These" instead of "This"
Corrected

l. 100-101, better explain how you substitute distance by time, and what the consequences are, i.e. what is the minimum velocity of debris flows and – using a reference – how realistic that is
The procedure is described in l. 105 and l. 110. A reference has been added at line 105.

l. 108, the threshold of a ratio of 6 is arbitrary. Ideally, you provide some results on the quality loss for higher or lower thresholds. However, if that seems too much effort, at least write that you arbitrarily chose that specific threshold.
This aspect is already mentioned at l.107 - "Analyses of the seismic data of twelve events..."

l. 110, what is "lowes"?
It was an error, now corrected to "lowest"

l. 114, 0.8 is another arbitrary threshold. See suggestion above.
This threshold was selected by a trial and error procedure. We added this information in the text.

l. 122, "determined manually", this needs more detail. Based on which criteria?
We added "..based on the signal shape."

Table 2, "internal report" needs more detail. Which institution? Which date? Ideally, a name or DOI.
We now provide a clear reference for this report

l. 147, explain briefly why the velocities might be overestimated
We have added a short sentence

l. 185, not sure that "benefited" is a useful word in this context
We think it is, as our work took advantage of unique field datasets from diverse environments

l. 185, "different distances between the geophones", this stands in direct contrast to the statement made in line 129. Please clarify.

We clarify that  here the distance between two geophones is of concern, in contrast to the transversal distance from channel banks to geophones mentioned earlier at line 129.

l. 204, double "the"

Corrected

l. 246, Sentence seems to have a fragment. At least it does not make sense when reading it.

True, there was a mistake.

 We have rephrased it and simplified it.

There are numerous further grammatical glitches which I hope Copernicus' editorial and type setting team will spot and fix.